# Proteome allocations change linearly with the specific growth rate of *Saccharomyces cerevisiae* under glucose limitation

Jianye Xia [1,2,3], Benjamin J. Sánchez[2], Yu Chen [1,2], Kate Campbell[2], Sergo Kasvandik [4] & Jens Nielsen [2,5 ✉]

*Saccharomyces cerevisiae* is a widely used cell factory; therefore, it is important to understand how it organizes key functional parts when cultured under different conditions. Here, we perform a multiomics analysis of *S. cerevisiae* by culturing the strain with a wide range of specific growth rates using glucose as the sole limiting nutrient. Under these different conditions, we measure the absolute transcriptome, the absolute proteome, the phosphoproteome, and the metabolome. Most functional protein groups show a linear dependence on the specific growth rate. Proteins engaged in translation show a perfect linear increase with the specific growth rate, while glycolysis and chaperone proteins show a linear decrease under respiratory conditions. Glycolytic enzymes and chaperones, however, show decreased phosphorylation with increasing specific growth rates; at the same time, an overall increased flux through these pathways is observed. Further analysis show that even though mRNA levels do not correlate with protein levels for all individual genes, the transcriptome level of functional groups correlates very well with its corresponding proteome. Finally, using enzyme-constrained genome-scale modeling, we find that enzyme usage plays an important role in controlling flux in amino acid biosynthesis.

[1] State Key Laboratory of Bioreactor Engineering, East China University of Science and Technology, Shanghai 200237, China. [2] Department of Biology and Biological Engineering, Chalmers University of Technology, SE41296 Gothenburg, Sweden. [3] Key Laboratory of Systems Microbial Biotechnology, Tianjin Institute of Industrial Biotechnology, Chinese Academy of Sciences, Tianjin 300308, China. [4] Institute of Technology, University of Tartu, 50411 Tartu, Estonia. [5] BioInnovation Institute, Ole Maaløes Vej 3, DK2200 Copenhagen, Denmark. ✉email: nielsenj@chalmers.se

Saccharomyces cerevisiae is a widely used cell factory due to its robustness under industrial conditions[1]. It has therefore been engineered for the production of a range of different chemicals, such as isoprenoids[2], free fatty acids[3], biopharmaceutical proteins[4] and many precursors of high-value-added products[5]. As one of the most studied eukaryal cells, S. cerevisiae is also extensively used as a model organism for deciphering molecular mechanisms in cellular and molecular biology[6–9]. However, insight into how yeast cells coordinate their resources when grown at different growth rates, especially on the allocation of their proteome, is lacking. In addition, what determines protein resource distribution across different functional groups also remains unclear[10].

Cellular proteins are responsible for all key cellular functions, i.e., transcription (RNA polymerases), catalyzing reactions (enzymes), transporting molecules across membranes (transporters), translation (ribosome), folding and assembly of proteins (chaperones), signal transduction (kinase or phosphatase), and many other functions[11]. Cellular phenotypes under different conditions are therefore determined by the fine tuning of proteome fractions across different functional groups. Within the context of industrial strain engineering, understanding the underlying principles for proteome allocation at different cell growth rates can therefore be of importance for both strain engineering and bioprocess optimization[12]. Metzel-Raz et al.[10] investigated proteome allocation across a wide range of specific growth rates (0.07–0.4 h$^{-1}$) under different conditions (nitrogen, phosphate, and carbon limitation) for S. cerevisiae and found condition-dependent proteome profiling and a strong positive linear relationship between the fraction of translational proteins and specific growth rate. Similar results have been found for E. coli[13], and an elegant equation called the bacterial growth law describing the relation between specific growth rate and the ribosome plus nonribosome fractions was proposed[14]. Although S. cerevisiae showed varying proteome allocation patterns when cultured under different conditions (five different nutrient limitation experiments[15]), a thorough investigation of the proteome allocation of S. cerevisiae under carbon limitations is lacking.

It is well known that S. cerevisiae shows distinct phenotypes when cultured aerobically under limited or excess glucose, first confirmed by De Deken[16] and named after the English biochemist Herbert Grace Crabtree, who first observed these phenomena in tumor cells[17]. Using a proteome-constrained flux balance analysis study with S. cerevisiae, it was recently shown that the Crabtree effect can be explained by a trade-off between fermentative pathway enzymes and oxidative phosphorylation enzymes[18] and that glycolytic enzymes have a much higher ATP synthesis capacity per enzyme mass than respiratory enzymes. However, there has been no thorough systematic study involving the analysis of different omics levels in S. cerevisiae across a wide range of specific growth rates where the Crabtree effect sets in.

Here, we carried out a multiomics study of S. cerevisiae in glucose-limited chemostats across a wide range of specific growth rates (equal to dilution rates under steady state) ranging from 0.025 to 0.4 h$^{-1}$. To understand the proteome allocation pattern when yeast cells increase their specific growth rate and the underlying principles that regulate the process, we performed absolute proteome analysis and integrated both transcriptome and phosphoproteome data to interpret protein resource allocation patterns at different specific growth rates. Finally, a quantitative proteome-constrained genome-scale metabolic model was used to investigate enzyme usage across the whole metabolic network of S. cerevisiae under a wide range of specific growth rate conditions. Together with metabolome data of amino acid biosynthetic pathways, the model gave a detailed interpretation of

how flux is controlled through these pathways with increasing specific growth rates.

## Results

**Linear relationships between functional categories of the proteome and cell-specific growth rate.** S. cerevisiae was cultured at nine different specific growth rates (from 0.025 to 0.4 h$^{-1}$, Fig. 1A) in biological triplicate, and all major exchange fluxes were quantified (Fig. 1B). Two distinct phenotypes were clearly observed across the studied dilution rate range. Although the dissolved oxygen (DO) level remained above 40% saturation, ethanol was produced under aerobic conditions when the specific growth rate increased above 0.28 h$^{-1}$ (Fig. 1B), a well-known phenomenon referred to as the Crabtree effect[16,19]. This also resulted in an increasing respiratory quotient (RQ), i.e., carbon dioxide production relative to oxygen consumption (Fig. 1B). However, we find that the decoupling of the specific oxygen uptake rate ($q_{O_2}$) and specific carbon dioxide production rate ($q_{CO_2}$) does not occur at the same specific growth rate when ethanol begins to accumulate, as decoupling occurs at dilution rates below 0.28 h$^{-1}$ (which is believed to be the critical dilution rate that triggers the Crabtree effect). Similar results were also observed for another yeast strain, S. cerevisiae CBS 8066[20]. This inconsistency has not been addressed before; moreover, we observed similar inconsistencies in the allocation pattern changes of different proteome functional groups at dilution rates below 0.28 h$^{-1}$ (Fig. 1C, D). S. cerevisiae relies on oxidative phosphorylation as the dominant route for ATP production when the dilution rate is below 0.28 h$^{-1}$, known as the purely respiratory condition. For dilution rates above 0.28 h$^{-1}$, cells alter their metabolism to increasingly use substrate-level phosphorylation for the production of ATP, known as the respirofermentative condition.

To investigate how S. cerevisiae allocates its proteome at different specific growth rates, absolute proteome measurements under the 9 glucose-limited chemostats were performed, which resulted in quantitative measurements of 1787 identified proteins (Supplementary Data 1 and Supplementary Note 1 for more details about the proteins identified under different conditions) that were grouped into 11 categories according to their physiological functions (adapted and modified accordingly from ref. [10], details of each category composition can be found in Supplementary Data 2). We found that proteome allocation varied with the specific growth rate. As protein constitutes a large part of the dry cell weight (ranging from 30 to 50% (w/w) in S. cerevisiae[21]), the protein synthesis (i.e., translation by ribosome) rate is confidently anticipated to be a determinant of the specific growth rate[22]. Our proteome data also confirmed this finding. By correlating the fraction of proteins engaged in translation with specific growth rates, we found a perfect linear relationship (with a Pearson correlation coefficient $R = 1$) (Fig. 1C). The correlated relationship is shown in Eq. (1):

$$f_r = 0.35\mu + 0.13 \tag{1}$$

where $f_r$ denotes the ribosome protein fraction and $\mu$ is the specific growth rate. The linear relationship between the ribosome protein fraction and $\mu$ has been confirmed in different species (both yeast[10] and E. coli[13]) under various conditions (N limitation, C limitation, P limitation[10] and with or without amino acids in the medium[23]) and has been referred to as the bacterial growth law in prokaryotes[14]. From our proteome data, we calculated the average translation rate of the ribosomes in our strain to be ~10 amino acids/ribosome/s (details of the calculation can be found in Supplementary Note 2 and Supplementary

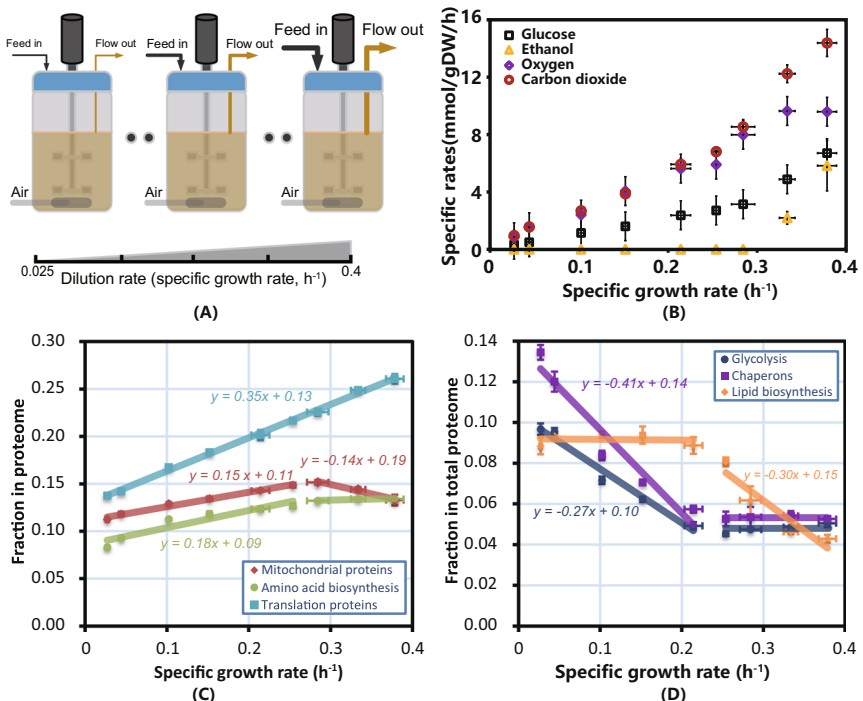

**Fig. 1 Experimental design, chemostat series experimental results and linear relationship between different protein categories and specific growth rates. A** The experimental design was based on glucose-limited chemostat cultures operated within a range of dilution rates (equal to specific growth rates, nine different values) from 0.025 to ~0.4 h$^{-1}$. The line width of the arrow represents the flow of feed and effluent. **B** Key exchange fluxes measured at different dilution rates. All error bars were obtained based on biological triplicates and data are presented as mean ± SD. **C** Changes in the fraction of the proteome allocated to translation (ribosome protein), mitochondrial, and amino acid biosynthesis across different specific growth rates. Protein allocation in these regions is correlated with specific growth rates. Error bars derived from $n = 3$ biologically independent samples and data are presented as mean ± SD. **D** Changes in the fraction of the proteome allocated to chaperones, glycolysis, and lipid biosynthesis across different specific growth rates. Error bars derived from $n = 3$ biologically independent samples and data are presented as mean ± SD. Protein allocation in these categories is inversely correlated with the specific growth rate. Source data are provided as a Source Data file.

Data 4), which agrees well with what has been reported earlier for *S. cerevisiae*[24,25].

Two other groups (amino acid biosynthesis and mitochondrial proteins) also showed an increasing tendency along with the specific growth rate (Fig. 1C). Unlike the translation proteins, these two groups showed a metabolism-dependent profile (Fig. 1C). Under respiratory conditions, the proteome fractions for these two groups increased linearly, while the profile changed when the metabolism changed to respirofermentative conditions ($\mu > 0.28\,h^{-1}$). The proteome fraction for mitochondrial proteins decreased with a slope of −0.14 under respirofermentative conditions. This is consistent with the hypothesis that the Crabtree effect is caused by a trade-off of enzyme mass invested into the two ATP-supplying pathways[18]: substrate-level phosphorylation (mainly relying on glycolytic enzymes to supply ATP) and oxidative phosphorylation (mainly relying on the electron transfer chain enzyme complexes located in the mitochondria). In contrast, the proteome fraction for amino acid biosynthesis reached and was maintained at a constant level (13% of the total proteome) when metabolism shifted from purely respiratory to respirofermentative metabolism. A recent publication showed that supplementing minimal medium with amino acids can increase the *S. cerevisiae* growth rate[23], which revealed that relieving the demand of amino acid biosynthetic enzymes can support higher proteome resource allocation to translational proteins.

To accommodate an increased allocation of proteome fraction for amino acid biosynthesis and protein translation (mainly ribosome), the cell needs to decrease the fraction of proteome allocated to other proteome categories. These were also found to

be highly metabolism dependent (Fig. 1D). Two distinct conditions are shown that coincide with the cell's shift in its respiration properties. Under respiratory conditions (strictly speaking, under conditions where $q_{O2}$ and $q_{CO2}$ are coupled and in which RQ equals 1), the cell decreased its proteome fractions for chaperones and glycolysis to support the increasing demand of the proteome for translation, amino acid biosynthesis and mitochondrial proteins (Fig. 1D), while under conditions where $q_{O2}$ and $q_{CO2}$ are decoupled (RQ > 1), the fractions of these two parts stopped decreasing but were kept at a constant level. The fraction of proteins engaged in lipid metabolism remained at a constant level under conditions where chaperone and glycolysis allocation decreased, but this fraction decreased linearly when $q_{O2}$ and $q_{CO2}$ were decoupled (where the glycolysis and chaperone fractions reached their minimal value). Comparing Fig. 1C, D shows that the proteome allocation pattern changed significantly before yeast reached its critical specific growth rate ($0.28\,h^{-1}$), which triggered respirofermentative metabolism and the accumulation of ethanol.

**Changes in the proteome allocation pattern with growth rate are accompanied by the same allocation pattern of the transcriptome at the functional group level.** To understand the observed changes in the proteome allocation of *S. cerevisiae*, we performed absolute transcriptome analysis under the same conditions. The number of mRNAs identified was much larger than the number of proteins, with 5401 transcripts and 2821 proteins identified in total under all conditions,. However, the overall mRNA abundance was much lower than the overall protein abundance (Supplementary Fig. 2), as the most abundant (from

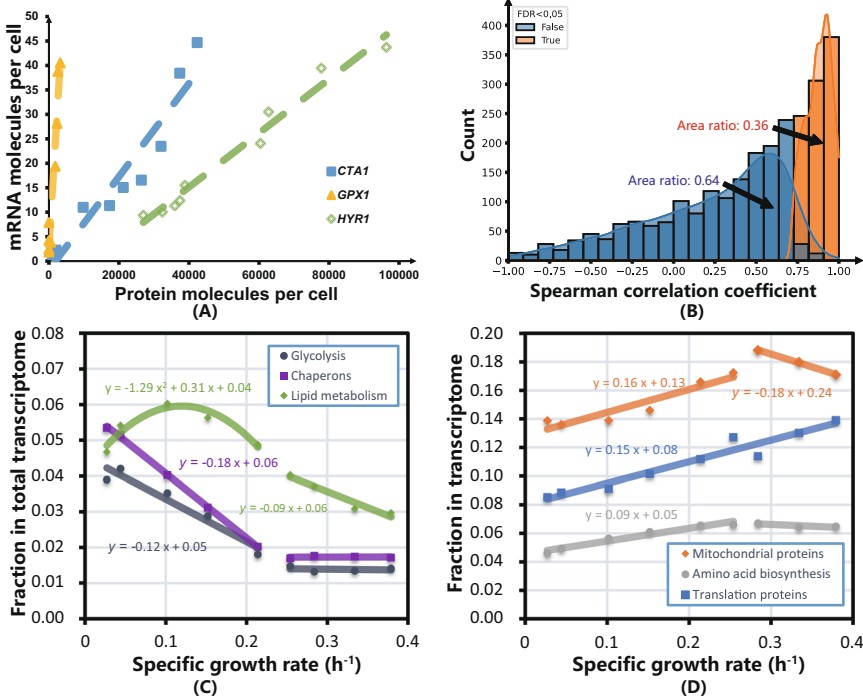

**Fig. 2 Correlation analysis of the transcriptome and proteome and allocation of the transcriptome into different functional categories. A** Perfect monotone (with Spearman correlation coefficient $R = 1$) correlations between absolute protein and mRNA abundance at the individual gene level across different growth rates, illustrated with the three genes *CTA1*, *GPX1*, and *HYR1*. These three genes are chosen here since they stand out when performing GO term enrichment analysis and commonly fall into three GO function terms. **B** Distribution of Spearman rank correlation coefficients for individual proteins vs. their corresponding mRNA levels across different specific growth rates. Here, FDR < 0.05 was used to distinguish between significant correlation (in orange) and random correlation (in blue). **C** Changes in the fraction of mRNAs allocated to translation, mitochondrial, and amino acid biosynthesis across different specific growth rates. **D** Changes in the fraction of mRNAs allocated to chaperones, glycolysis, and lipid metabolism across different specific growth rates. Source data are provided as a Source Data file.

10 to 90% percentile) mRNA ranged from 2 to 25 copies per cell, whereas the most abundant (from percentile 10 to 90%) protein ranged from 504 to 67,440 molecules per cell. The ratio of the corresponding protein to its coding mRNA covers a wide range, i.e., from 0.7 to 27,424 proteins per mRNA molecule. Detailed abundance information for all mRNAs across the nine dilution rates can be found in Supplementary Note 3 and Supplementary Data 3.

A linear regression between the log value of mRNA and protein abundance for all data points together (Supplementary Fig. 3) resulted in a linear regression coefficient of determination $R^2 = 0.52$, which is consistent with previous publications[26,27]. However, we noticed that neither the mRNA nor the protein abundance data were normally distributed, so we used Spearman rank correlation instead of Pearson correlation for individual mRNA-protein pairs to infer the relationship between protein and mRNA. Even though 54 mRNA-protein pairs showed perfect monotone correlation (with a Spearman correlation coefficient of 1), such as for the genes *CTA1*, *GPX1*, and *HYR1* (Fig. 2A), fewer mRNA-protein pairs (932 among 2568 mRNA-protein pairs, less than 40% of all pairs) showed statistically significant positive correlations with an false discovery rate (FDR) < 0.05 (Fig. 2B). An earlier study[28] also showed that mRNA abundance alone can explain only 40% of corresponding protein levels, but it explains over 85% of corresponding protein abundance when posttranscriptional and translational effects are considered. It should be noted that the slopes of the regressed lines between protein and mRNA abundance vary among different transcripts (Supplementary Note 4 and Supplementary Fig. 4), indicating variations in the translation rate of individual transcripts. Similar results have also been reported by ref. [29]. By using in vivo visualization of

single mRNA molecule translations, Yan et al.[30] also reported heterogeneity in the translation of individual mRNAs in the same strain. GO term enrichment analysis (Supplementary Fig. 5) revealed that for ribosome-related genes, the mRNA level and protein level were significantly correlated (correct $p$ value < 0.01).

Even though a poor correlation between mRNA and protein was observed at the individual gene level for ~60% of the measured proteins, we did observe a correlation between transcriptome fractions and specific growth rate at the functional group level, as in the case of the proteome (Supplementary Fig. 6). We divided the transcriptome into the same 11 functional groups and found that fractions of mRNAs allocated to the same functional groups showed similar patterns of change with specific growth rates to those of the proteome, except for lipid metabolism-related transcripts, which showed a different allocation pattern at low specific growth rates (Fig. 2C, D). These results implied that *S. cerevisiae* seems to allocate both its transcriptome and proteome functional groups with a similar pattern under glucose-limited steady-state conditions. One notable difference between mRNA allocation and proteome allocation, however, is the fraction allocated to mitochondrial genes, which was much larger in the transcriptome than in the proteome.

The overall consistent allocation pattern for both the transcriptome and proteome at the functional group level showed that *S. cerevisiae* allocates its resources based on gene function at both the transcriptome and proteome levels. A Pearson correlation between transcriptome group fractions and proteome group fractions was carried out on each functional group and showed that all six groups shown in Figs. 1C, D and 2C, D were significantly correlated (Supplementary Fig. 6 and Supplementary

Table 4). This may indicate that *S. cerevisiae* organizes its resources according to functional groups at both the transcriptome and proteome levels. However, how the cell regulates these resources under different specific growth rates still needs further investigation.

**Protein phosphorylation likely affects the majority of glycolysis and chaperone protein activities.** Next, we analyzed correlations between reaction flux and mRNA or protein abundance. The genome-scale metabolic model Yeast7.6 was used together with measured exchange fluxes, i.e., $q_{Gluc}$, $q_{Ethanol}$, $q_{O_2}$, $q_{CO_2}$, and $\mu$, to estimate all 2302 fluxes in the metabolic network at the nine different dilution rates. A total of 318 fluxes were paired with both a detected transcript and a detected protein level, and these pairs were used for correlation analysis across the nine dilution rates. A scatter plot of the Pearson correlation coefficients for both mRNA vs. flux and protein vs. flux is shown in Fig. 3A. The correlation coefficient distribution for both pairs is plotted at the margin of the main plot in Fig. 3A. We filtered out significantly ($p$ value corrected with FDR < 0.05) positive (red points) and negative correlations (blue points) from other points that were not significantly correlated for both mRNA-flux and protein-flux pairs. To clearly view where these reactions are located in the reaction network, we then mapped all reactions that were significantly correlated (red and blue points in Fig. 3A) with protein levels on the reaction network of the yeast cell using the iPath3 tool[31]. The results are shown in Supplementary Fig. 15. Pathway enrichment analysis was carried out for both significantly positively correlated and significantly negatively correlated reactions (shown in Fig. 3B). From this analysis, we observed that among the central carbon metabolism pathways, reactions in the TCA cycle and pathways that engaged in nucleotide biosynthesis (purine metabolism) showed positive correlations, while EMP pathway reactions mainly showed negative correlations.

To replenish proteins for translation, mitochondria and amino acid biosynthesis when cell growth becomes faster, the cell decreases its investments into proteins involved in glycolysis and lipid biosynthesis as well as chaperones. This observation holds not only in terms of relative abundance but also in terms of the absolute abundance levels of both mRNA and proteins in each functional group (Supplementary Fig. 7). It is interesting that the abundances of both glycolytic enzymes and chaperones decrease as the cell grows faster under respiratory conditions. This is surprising because the glycolytic flux increases with increasing growth rate (Fig. 3C and Supplementary Fig. 9D). This means that the specific activity of the pathway enzymes or the usage of the enzymes increases with the growth rate. The same situation was observed for chaperones, even though there is an increased need for their activity during faster growth because of the increased protein synthesis. Thus, increased specific activity is also expected for chaperones. We were therefore interested in determining whether there was any factor associated with the increased specific activity of both chaperones and glycolytic enzymes. For this purpose, we checked both the metabolome and phosphoproteome to investigate whether metabolite allocation regulation or phosphorylation regulation took place.

We performed Bayesian inference to check the metabolite allosteric effects on reactions of glycolysis and amino acid biosynthesis pathways, following the method used by ref. [15]. However, here, a general loglinear kinetics model derived from thermodynamics was used instead of Michaelis–Menten kinetics. An intrinsic turnover number concept (see Methods section for details) was proposed in this model that describes the allosteric effect of each metabolite on the reaction enzyme. The proposed intrinsic turnover number indicates how the corresponding metabolite regulates enzyme activity, with a positive value indicating activation and a negative value indicating inhibition. The posterior distribution of the intrinsic turnover number for all tested metabolites was obtained by using Markov chain Monte–Carlo-based Bayesian inference. However, only a positive effect of ATP on phosphoglycerate kinase, NADH on glyceraldehyde 3-phosphate dehydrogenase and CIT on citrate synthase were strongly indicated by our Bayesian inference results. Compared to the clear linear decrease in the glycolysis pathway enzyme fractions along with the specific growth rate, we did not observe the effects of allosteric regulation on the regulation of enzyme-specific activity, at least for the EMP pathway, as analyzed here. More detailed results of the Bayesian inference can be found in Supplementary Note 5.

Phosphorylation of proteins is a reversible posttranslational modification of amino acid residues (serine, threonine, or tyrosine); introducing a covalently bound phosphate group to the protein molecule alters the structural conformation of the protein, thereby activating or deactivating the protein or modifying its function[32]. In yeast, more than half of the ~900 metabolic enzymes have phosphorylation sites[33,34]. In our phosphoproteomics data, 5971 phosphopeptides associated with 1450 proteins were identified, and among them, 1761 phosphopeptides associated with 782 proteins showed increasing or decreasing trends with the specific growth rate. An enrichment analysis of these proteins was carried out with respect to GO terms or KEGG pathways (Supplementary Fig. 10).

Many phosphorylation sites were found in glycolytic enzymes, i.e., phosphorylation site(s) were detected in all such enzymes except Pgi1 and Cdc19. Seven of the ten enzymes of this pathway showed a linear decrease in phosphorylation level as the cell grew faster under respiratory conditions (Supplementary Fig. 8). Using an algorithm for inferring functional phosphorylation events (FPE)[33], we found that most of the phosphosites on the seven enzymes could inhibit their activities (Supplementary Fig. 8). Previously, FPE on Pfk2 S163 (inhibiting enzyme activity) and Fba1S313 (activating enzyme activity) were reported by ref. [35]. Our data confirmed the S163 phosphorylation inhibition effect, but we inferred no significant effect on S313 of Fba1. In addition, an activating effect on S160 of Pfk2, as well as inhibitory effects on Y90 of Gpm1, on T178, S179, S185, S188, S189, S192 of Pfk1, on S149 of Tdh1, Tdh2 and Tdh3, on S149 of Pgk1, on Y75 of Pgk1, and on Y259 of both Eno1 and Eno2 were also observed (Supplementary Fig. 8).

For chaperones, the algorithm used to propose FPEs could not be used, as it was not possible to obtain reaction flux information for these proteins. We therefore analyzed the phosphorylation levels for all chaperones, and among the 48 chaperones identified in yeast, 43 phosphorylated peptides were detected in 17 chaperones, and phosphorylation levels for 30 of the 43 phosphorylated peptides (69.8%) showed a decreasing trend with increasing specific growth rate. Even though only 35% (17 over 48) of the chaperones showed a decreasing phosphorylation level, these chaperones accounted for more than 80% of the mass fraction of the chaperones at the lowest specific growth rate of $0.025\,h^{-1}$ and 58% at the highest specific growth rate of $0.38\,h^{-1}$ (Supplementary Fig. 11).

Based on these results, we propose that phosphorylation of both glycolytic enzymes and chaperones may be why these proteins showed lower activities with higher protein abundance at low specific growth rates of *S. cerevisiae*. To validate this hypothesis, further experimental efforts are needed to clarify the roles of these aforementioned phosphorylation sites.

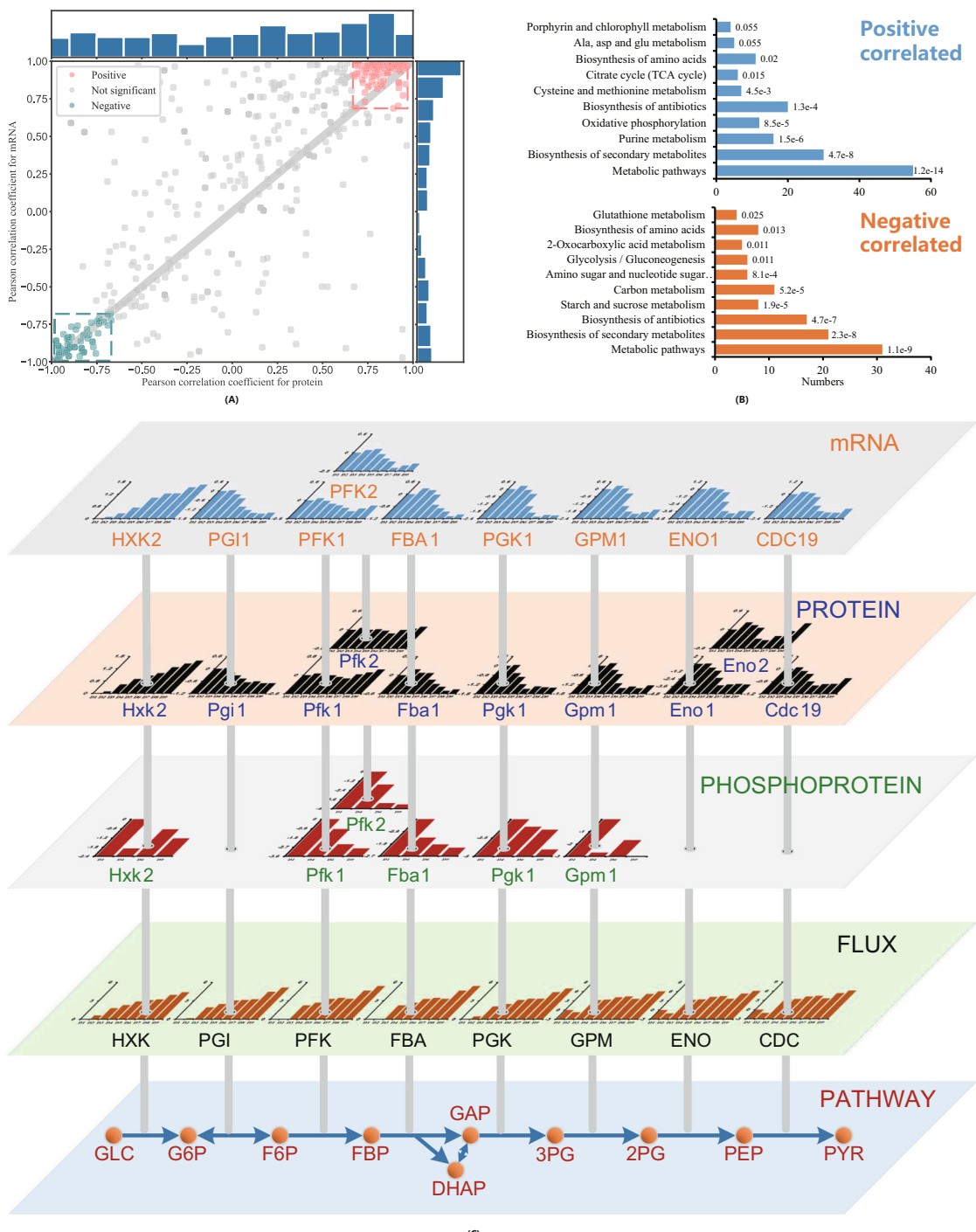

**Fig. 3 Correlation analysis between reaction fluxes and abundances of the corresponding mRNA and protein. A** Pearson correlation coefficients for both mRNA/flux and protein/flux pairs, with the distribution of Pearson correlation coefficients for protein/flux pairs shown on the right margin and for mRNA/flux pairs shown on the top margin. **B** Pathway enrichment analysis for both significantly positively correlated and negatively correlated reactions. Figures above each bar are adjusted *p* values. **C** Overview of EMP pathway reactions with changes in absolute reaction fluxes, protein phosphorylation levels, absolute protein abundance and absolute mRNA abundance. The abscissas for all bar plots are specific growth rates of yeast cells. Details of the bar plots included in this panel can be found in the Supplementary Information. Source data are provided as a Source Data file.

**Enzyme saturation plays an important role in fulfilling amino acid biosynthesis demand as yeast cells grow faster.** Hierarchical regulation analysis can be used to decipher at which level a reaction is regulated under both steady state and dynamic conditions[36]. Flux regulation contributions of enzymes, metabolites, transcripts, or posttranslation modifications can be determined by hierarchical regulation coefficients $\rho_i$, with $i$ standing

for e (enzyme regulation coefficient), m (metabolite regulation coefficient), t (transcript regulation coefficient), or p (protein phosphorylation regulation coefficient). According to the analysis by ref. [33], when $0.5 < \rho_e < 1.5$, the reaction flux is regulated mainly by enzyme abundance. Pairwise hierarchical analysis, taking the slowest specific growth rate $0.027\,h^{-1}$ as the reference, was used to calculate $\rho_e$ based on the proteome and fluxome data with both

the central carbon metabolism and amino acid biosynthesis pathways (Supplementary Fig. 12). The results show scarce protein abundance regulation on the reaction fluxes of these pathways. The combined results of protein phosphorylation analysis and hierarchical regulation analysis for the EMP pathway indicate that the activity of the majority of enzymes of this pathway may be under regulation by phosphorylation modification, with dephosphorylation seeming to activate enzymes when the enzyme abundance is low at high specific growth rates, as discussed above.

However, we generally did not find many protein phosphorylation events related to the biosynthesis of amino acids and nucleotides (Supplementary Fig. 10), and it is unclear what regulates flux through these pathways. To analyze this, we used a Genome-scale metabolic model with Enzymatic Constraints using Kinetic and Omics data (GECKO)[37], which uses the catalytic capacity constraint of each individual enzyme ($k_{cat}$ multiplied by the enzyme concentration) together with the stoichiometric constraints normally used in flux balance analysis. GECKO allows improved calculation of fluxes through the metabolic network of yeast compared with regular flux balance analysis, and it further allows calculation of the enzyme requirement for each reaction. The latter was used to calculate enzyme usage: the model calculated the enzyme requirement divided by the corresponding measured enzyme concentration (molecules per cell). As there were few phosphorylation events of biosynthetic enzymes, it is a fair assumption to use enzyme usage as a proxy for enzyme saturation. Using this method, we calculated nonzero usages of 321 enzymes and found 58% of them to have less than 10% usage on average but 11% of them to have >90% usage on average (Supplementary Fig. 13). Furthermore, for 180 of the enzymes (56%), their usage showed a positive correlation with the specific growth rate (with Pearson correlation coefficient $R > 0.8$). The top ten enzymes with the strongest correlation (with Pearson correlation coefficient $R > 0.99$) between enzyme usage and specific growth rate were all enzymes involved in synthesizing the main building blocks of the cell (Fig. 4A, mostly amino acids).

All 180 enzymes showing a strong positive correlation of usage with specific growth rate were analyzed for KEGG pathway overrepresentation (Fig. 4B). Here, translation (in the form of aminoacyl-tRNA biosynthesis) and the synthesis of building blocks (mainly amino acids) show consistent trends of high correlation (Fisher exact test $p$ value $< 0.01$), indicating increased saturation of enzymes involved in biosynthesis at increasing specific growth rates. We also found that increased enzyme usage for glycolysis occurs above the critical dilution rate ($0.28\,h^{-1}$), i.e., only during respirofermentative metabolism (Fig. 4C). This makes sense, as almost no phosphorylation effects on this metabolic regime were observed (Supplementary Fig. 9C).

To further evaluate whether the metabolomics data support the model prediction of increased enzyme saturation with growth rate in amino acid biosynthesis, we checked the relative abundance of the intermediates in the amino acid biosynthetic pathways. According to the SGD database (https://www.yeastgenome.org/), there are 59 intermediates for all amino acid biosynthetic pathways, out of which 20 were measured by our metabolome analysis. Fourteen of these 20 intermediates showed a positive correlation ($R > 0.8$) with the specific growth rate (Fig. 4D). Specifically, we found four out of the five measured intermediates of the arginine pathway and four out of the seven measured intermediates of the lysine pathway to have a positive correlation with growth rate (this is consistent with the enzyme usage results for the lysine biosynthesis pathway, Fig. 4B). Arginine and lysine are both derived from 2-oxoglutarate, and their biosynthesis is characterized by low levels of metabolite/feedback regulation[38].

For another six amino acids (Cys, Trp, Val, Met, Leu, His), only one biosynthetic intermediate had a concentration strongly correlated with the specific growth rate.

Fourteen amino acids (Ala, Arg, Asn, Cys, Gln, Gly, His, Ile, Leu, Met, Phe, Pro, Trp, Tyr), however, themselves showed a negative correlation with the specific growth rate under purely respiratory conditions (Supplementary Fig. 14). Except for Ala, Asn, Cys and His, the biosynthesis of these amino acids is characterized by extensive metabolite regulation and strong feedback inhibition at the amino acid level. The decreasing concentration of these amino acids seems to contradict the increasing enzyme usage for aminoacyl-tRNA biosynthesis (Fig. 4B). However, the abundance of tRNAs has been reported previously to increase with specific growth rate[39,40], which could explain the increased enzyme usage of these reactions. Furthermore, the $K_m$ values for amino acids in aminoacyl-tRNA biosynthesis are reported to be very low (in the μM range)[41] compared with the intracellular amino acid concentrations (in the mM range). We then performed a correlation analysis between reaction flux and individual aminoacyl-tRNA synthase (see Supplementary Table 5), and the results indicated that aminoacyl-tRNA biosynthesis could be controlled by the tRNA synthase protein level. Flux toward amino acids in aminoacyl-tRNA biosynthesis would therefore be relatively insensitive to changes in the intracellular amino acid concentration. The decreasing concentration of amino acids with a specific growth rate therefore seems to occur to relieve feedback inhibition and thereby allow increased flux through the amino acid biosynthetic pathways. Increased enzyme concentration and enzyme saturation in the amino acid metabolism further supports this flux increase for increasing growth rates.

## Discussion

Based on transcriptome, proteome, and metabolome data from a series of glucose limitation chemostat steady states covering both respiratory and respirofermentative metabolism conditions, we studied how yeast regulates its proteome allocation at different specific growth rates. Simple linear relationships were observed between several functional protein groups and the specific growth rate. The group of translation-related proteins showed a perfect linear increase with the cell-specific growth rate over the whole range of specific growth rates studied, which has been referred to in bacteria as the growth law[14], first found in *E. coli*; a similar relationship has been confirmed for *S. cerevisiae*[10]. Here, we also found linear relationships for other functional protein groups.

Our transcriptome and proteome data showed similar allocation patterns for proteins and mRNAs in functional groups. Even though there is a poor linear relationship between mRNA and proteins in general due to the vast range of protein/mRNA ratios among different genes, the mRNA levels were positively correlated with protein levels within the same functional group (Supplementary Fig. 6). This indicates that some underlying mechanisms regulate the expression and translation of functional groups, regardless of the translation efficiency of individual genes.

Interestingly, we found a linear decrease in the abundance of both glycolytic enzymes and chaperones as the cell-specific growth rate increased under respiratory metabolism conditions, despite the need for increased activity of these proteins. Both hierarchical regulation analysis and FPE inference indicated that the increased glycolytic flux could be caused by the decrease in inhibitory phosphorylation events. An argument may be proposed that transcription factors may play an important role in causing the observed glycolysis flux regulation; however, detailed analysis of this provides no positive support (see Supplementary Note 6). Additionally, more than 80% of the chaperones (by

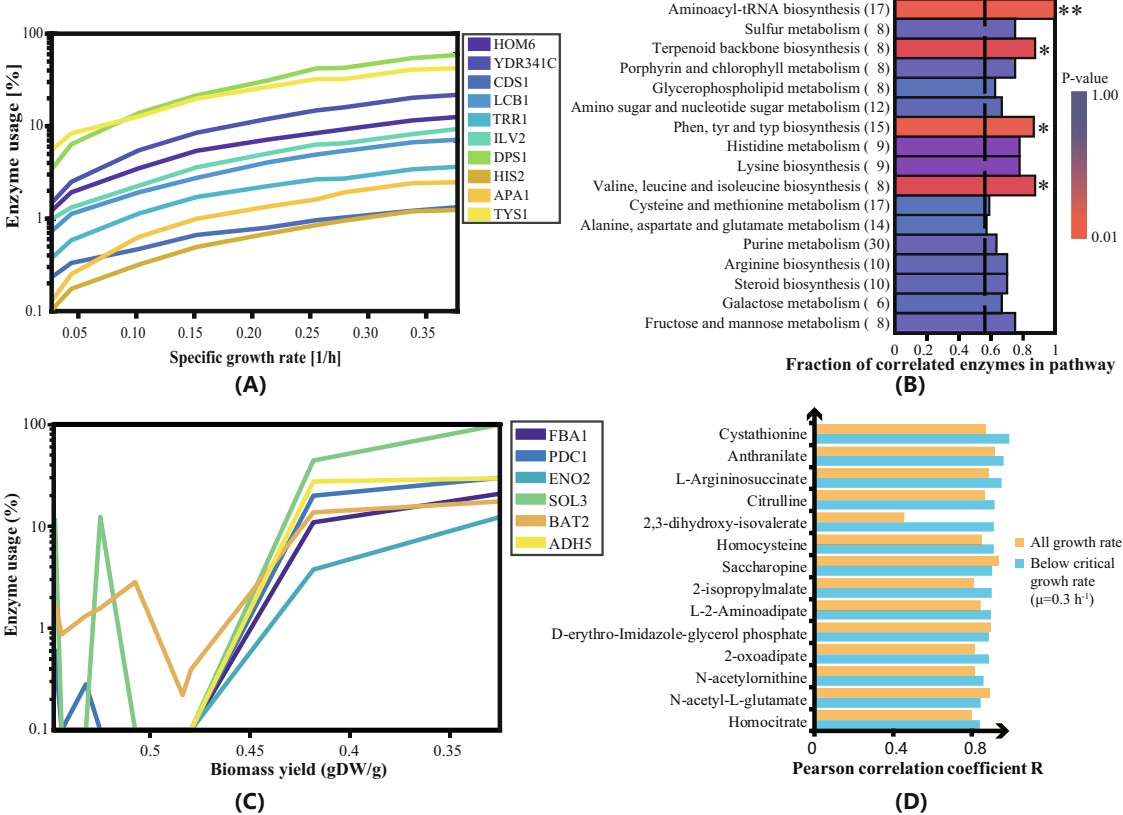

**Fig. 4 Enzyme saturation and correlation analysis for various metabolic pathways of *S. cerevisiae* with a range of specific growth rates covering both respiratory and respirofermentative metabolism. A** Enzyme usage of ten metabolic proteins as a function of specific growth rate (from ~0.025 to ~0.4 h$^{-1}$). **B** Metabolic pathway enrichment of enzymes correlated with the increasing specific growth rate. Values in parentheses are the number of enzymes in each KEGG pathway, and the *p* values were calculated by one-sided Fisher's test (asterisk * indicates a *p* value <0.1, ** indicates a *p* value <0.01). **C** Enzyme usage change trends with different biomass yields for glycolysis enzymes. **D** Correlation of the abundance of intermediates in the amino acid biosynthesis pathway and specific growth rate under respiratory (blue) and both respiratory and respirofermentative conditions (orange). Source data are provided as a Source Data file.

weight) showed decreased phosphorylation levels when the growth rate increased, again indicating possible activation of these proteins through dephosphorylation.

We also utilized metabolic modeling to quantitatively investigate the protein usage for all enzymes in the metabolic network, and the results showed that both enzyme abundance and saturation in amino acid biosynthesis pathways increased to fulfill the increasing demand for building blocks, in particular amino acids, with increasing specific growth rates. This analysis was confirmed by an analysis of pathway intermediates that showed increasing trends with growth rate. Interestingly, however, the levels of proteogenic amino acids decreased with the specific growth rate, most likely due to a requirement for reduced feedback inhibition of pathway enzymes.

Based on our analysis, we investigated the patterns of protein allocation for *S. cerevisiae* across a wide range of specific growth rates under glucose-limited conditions. Our results revealed the underlying principles of how yeast coordinates its proteome resources. We found this to be highly correlated with their transcriptome functional group, whereas posttranslational modifications, enzyme saturation and allosteric regulation (mainly for amino acid biosynthesis) play important roles in controlling metabolic fluxes. In addition to expanding our insight into fundamental metabolic regulation in eukaryotic organisms, these findings also offer the potential to optimize the production of high-value chemicals through future yeast engineering.

## Methods

**Strains used in this study**. Unless otherwise stated, the yeast *Saccharomyces cerevisiae* CEN.PK113-7D (MATα, MAL2-8c, SUC2) was used. For quantitative proteome analysis, the strain *S. cerevisiae* CEN.PK113-7D *lys1::kanMX* was used for obtaining $^{15}$N,$^{13}$C-lysine-labeled protein internal standard, which was constructed by ref. [26].

**Media and culturing methods**. Minimal mineral medium was used, which contained 10 g of glucose, 5 g of (NH$_4$)$_2$SO$_4$, 3 g of KH$_2$PO$_4$ and 0.5 g of MgSO$_4$ per liter, with 1 ml of trace metal solution and 1 ml of vitamin solution. The trace metal solution contained the following per liter: FeSO$_4$•7H$_2$O, 3 g; ZnSO$_4$•7H$_2$O, 4.5 g; CaCl$_2$•2H$_2$O, 4.5 g; MnCl$_2$•4H$_2$O, 1 g; CoCl$_2$•6H$_2$O, 300 mg; CuSO$_4$•5H$_2$O, 300 mg; Na$_2$MoO$_4$•2H$_2$O, 400 mg; H$_3$BO$_3$, 1 g; KI, 100 mg; and Na$_2$EDTA•2H$_2$0, 19 g (pH = 4). The vitamin solution contained the following per liter: d-biotin, 50 mg; 4-aminobenzoic acid, 0.2 g; Ca pantothenate, 1 g; pyridoxine-HCl, 1 g; thiamine-HCl, 1 g; nicotinic acid, 1 g; and myoinositol, 25 g (pH = 6.5). The chemostat feeding medium was the same as the minimal mineral medium except for the use of 7.5 g/l glucose was used instead of 10 g/l glucose.

To generate the internal standard for quantitative proteomics, the lysine auxotrophic strain (CEN.PK113-7D *lys1::kanMX*) was cultured with heavy labeled $^{15}$N,$^{13}$C-lysine (Cambridge Isotope Laboratories). Fully labeled biomass (>95% incorporation) was produced and harvested under four different stages of the culture process: Fed-batch cultures of the auxotrophic strain were carried out in three 1 l bioreactors with three exponential feeding rates, 0.1, 0.2, and 0.35 h$^{-1}$, and the feeding continued for at least one dilution volume before cell harvest. The harvest biomass samples were mixed together and thus comprised cells in the batch phase (Sample S1) and in each of the three exponential feeding phases (Sample S2 stands for sample 0.1 h$^{-1}$, Sample S3 for 0.2 h$^{-1}$ and Sample S4 for 0.35 h$^{-1}$). This was performed to collect biomass with varying proteome compositions, which would enable a broad spectrum of heavily labeled proteins in order to obtain as many quantifiable proteins as possible.

All remaining cultures were carried out under glucose-limited chemostat conditions with nine different dilution rates, ranging from 0.025 to ~0.4 h$^{-1}$, covering both respiratory (<0.28 h$^{-1}$) and respirofermentative metabolism (>0.28 h$^{-1}$). Experiments were carried out in 1 l bioreactors (Dasgip, Julich, Germany) equipped with an online off-gas analysis system alongside pH, temperature and DO sensors. An initial batch culture was carried out with inoculation of 10% seed cultures. The chemostat cultures were performed in 1 l bioreactors with a working volume of 0.5 l under aerobic conditions (DO > 40%) at 30 °C and pH 4.5. The continuous operation of chemostat cultures was performed at the end of the batch culture. To ensure that cells were growing at a steady state, chemostat cultures were run for at least five residence times before sampling.

**Sampling**. For extracellular metabolome measurements, broth was sampled and filtered immediately into 1.5 ml Eppendorf tubes and stored at −20 °C until high-performance liquid chromatography analysis was performed.

For transcriptome sample collection, 10 ml of broth was sampled and injected into a 50 ml falcon tube filled ~3/4 with ice. Cells were pelleted by centrifugation (2504 × $g$, 5 min, Centrifuge 5702R, Eppendorf, Germany), and biomass pellets were snap frozen in liquid nitrogen (N$_2$) and then transferred to a 1.5 ml Eppendorf tube and stored at −80 °C until further analysis.

For proteome sample collection, ~5 ml of culture broth was injected into a 50 ml preweighed falcon tube (prechilled on ice), and the tube was reweighed after sampling to determine the exact amount of broth collected. The samples were then pelleted by centrifugation (15,865 × $g$, 20 s, 4 °C, refrigerated centrifuge 4K15, Sigma, Germany), and the biomass pellet was washed in 1 ml of chilled PBS and then recentrifuged. Pellets were snap frozen in liquid N$_2$, transferred to a 1.5 ml Eppendorf tube and then stored at −80 °C until further analysis.

For intracellular metabolome sample collection, ~7 ml of culture broth was injected into a preweighed 50 ml falcon tube containing 35 ml of 40 °C 100% methanol. The tube was then reweighed after sampling to determine the exact amount of broth collected. The cells were pelleted in a precooled centrifuge (4000 × $g$, 3 min, −20 °C, refrigerated centrifuge 4K15, Sigma, Germany), the supernatant was discarded, and the samples were immediately stored at −80 °C until further analysis.

**mRNA sequencing**. Total RNA was extracted and purified using a Qiagen RNeasy Mini Kit, according to the user manual, with a DNase step included (Qiagen, Hilden, Germany). RNA integrity was verified using a 2100 Bioanalyzer (Agilent Technologies, Santa Clara, USA), and RNA concentration was determined using a NanoDrop 2000 (Thermo Scientific, Wilmington, USA).

To prepare RNA for sequencing, the Illumina TruSeq sample preparation kit v2 was used with poly-A selection. cDNA libraries were then loaded onto a high-output flow cell and sequenced on a NextSeq 500 platform (Illumina Inc., San Diego) with paired-end 2 × 75 nt length reads.

The raw data of reads generated by NextSeq 500 were processed using TopHat version 2.1.1[42] to map paired-end reads to the CEN.PK113-7D reference genome (http://cenpk.tudelft.nl/cgi-bin/gbrowse/cenpk/). Eight to 15 million reads were mapped to the reference genome with an average map rate of 95%. Cufflinks version 2.1.1[43] was then used to calculate the FPKM values for each sample. Mapped read counts were generated from SAM files using bedtools version 2.26.0[44]. Differential expression analysis was performed with the Bioconductor R package DESeq2[45].

**Absolute mRNA quantification**. To quantify RNA-Seq read counts, 18 mRNAs with FPKM values ranging from $3.4 \times 10^1$–$1.4 \times 10^4$ were selected, covering 80% of the dynamic range of mRNA expression under reference conditions ($D = 0.1$ h$^{-1}$). The absolute concentrations of these 18 mRNAs were then measured using the QuantiGene assay (Affymetrix, Santa Clara, CA, United States). Further details of this measurement can be found in our previous publication[26]. A positive linear correlation with a Pearson R value of 0.8 was achieved among these 18 selected mRNA absolute concentrations and their corresponding FPKM values (Supplementary Table 2). The same correlation was then applied to all remaining mRNAs identified by RNA sequencing to quantify their respective absolute mRNA levels. Absolute values of mRNA under other dilution rates were then calculated based on the fold-change obtained from differential expression analysis relative to the reference chemostat ($D = 0.1$ h$^{-1}$). Assuming that the weight of yeast cells does not change under different specific growth rates, a cell weight of 13 pg measured under reference conditions ($D = 0.1$ h$^{-1}$) was applied to all other chemostat conditions. The same assumption of a constant cell weight was applied for proteome absolute quantification. The calculated absolute mRNA concentrations were subsequently presented in the unit of [molecules/cell].

**Total and phosphoproteome sample preparation**. Cell pellets were resuspended in 10 volumes (relative to the cell pellet) of 6 M guanidine HCl, 100 mM Tris-HCl pH 8.0, and 20 mM DTT, heated at 95 °C for 10 min and sonicated with a Bioruptor (Diagenonde, Denville, NJ, United States) sonicator (15 min, "High" setting). Samples were further processed with FastPrep24 (MP Biomedicals, Santa Ana, CA, United States) twice at 4 m/s for 30 s with cooling between cycles. After removal of beads, the samples were precleared with centrifugation at 17,000 × $g$ for 10 min at

4 °C. After protein concentration measurement with a Micro-BCA assay (Thermo Fisher Scientific, Wilmington, USA), samples were spiked at a 1:1 ratio with the heavy lysine-labeled standard. For absolute quantification, 6 µg of heavy standard was spiked separately with 1.1 µg of UPS2 protein mix (Sigma Aldrich). Overall, 50 µg of protein was precipitated with a 2:1:3 (v/v/v) methanol:chloroform:water extraction. The precipitates were suspended in 7:2 M urea:thiourea and 100 mM ammonium bicarbonate. After disulfide reduction with 2.5 mM DTT and alkylation with 5 mM iodoacetamide, proteins were digested with 1:50 (enzyme to protein) Lys-C (Wako Pure Chemical Industries, Osaka, Japan) overnight at room temperature. The peptides were desalted using C18 material (3 M Empore) tips and reconstituted in 0.5% trifluoroacetic acid (TFA).

For the phosphoproteome analysis, cells were lysed as described above, except samples were not mixed with the heavy standard, and proteins were digested with dimethylated porcine trypsin (Sigma Aldrich, St. Louis, MO, United States) instead of Lys-C. Sample preparation was carried out as described by the EasyPhos protocol[46]. Five hundred micrograms of cellular protein was used as input for the phosphopeptide enrichment. Final samples were reconstituted in 0.5% TFA.

**Nano-LC/MS/MS analysis for protein quantification**. Two micrograms of peptides (for phosphoenriched samples, the entire sample) were injected into an Ultimate 3000 RSLC nano system (Dionex, Sunnyvale, CA, United States) using a C18 cartridge trap column in a backflush configuration and an in-house-packed (3 µm C18 particles, Dr Maisch, Ammerbuch, Germany) analytical 50 cm × 75 µm emitter column (New Objective, Woburn, MA, United States). Peptides were separated at 200 nl/min (for phosphopeptides: 250 nl/min) with a 5–40% B 240 and 480 min gradient for spiked and heavy standard samples, respectively. For phosphopeptides, a 90 min two-step separation gradient was used, consisting of 5–115% B for 60 min and 15–330% B for 30 min. Buffer B was 80% acetonitrile + 0.1% formic acid, and buffer A was 0.1% formic acid in water. Eluted peptides were sprayed onto a quadrupole-orbitrap Q Exactive Plus (Thermo Fisher Scientific, Waltham, MA, United States) tandem mass spectrometer (MS) using a nanoelectrospray ionization source and a spray voltage of 2.5 kV (liquid junction connection). The MS instrument was operated with a top-10 data-dependent MS/MS acquisition strategy. One 350–1400 $m/z$ MS scan (at a resolution setting of 70,000 at 200 $m/z$) was followed by MS/MS ($R = 17,500$ at 200 $m/z$) of the ten most intense ions using higher-energy collisional dissociation fragmentation (normalized collision energies of 26 and 27 for regular and phosphopeptides, respectively). For total proteome analysis, the MS and MS/MS ion target and injection time values were $3 \times 10^6$ (50 ms) and $5 \times 10^4$ (50 ms), respectively. For phosphopeptides, the MS and MS/MS ion target and injection time values were $1 \times 10^6$ (60 ms) and $2 \times 10^4$ (60 ms), respectively. The dynamic exclusion time was limited to 45 s, 70 s and 110 s for the phosphopeptide, spiked samples and heavy standard, respectively. Only charge states +2 to +6 were subjected to MS/MS, and for phosphopeptides, the fixed first mass was set to 95 $m/z$. The heavy standard was analyzed with three technical replicates, and all other samples were analyzed with a single technical replicate.

**Mass spectrometric raw data analysis and proteome quantification**. Raw data were identified and quantified with the MaxQuant 1.4.0.8 software package[47]. For heavy-spiked samples, the labeling state (multiplicity) was set to 2, and Lys8 was defined as the heavy label. Methionine oxidation, asparagine/glutamine deamidation and protein N-terminal acetylation were set as variable modifications, and cysteine carbamidomethylation was defined as a fixed modification. For phospho-analysis, serine/threonine phosphorylation was used as an additional variable modification. A search was performed against the UniProt (www.uniprot.org) *S. cerevisiae* S288C reference proteome database (version from July 2016) using the Lys-C/P (trypsin/P for phosphoproteomics) digestion rule. Only protein identifications with a minimum of 1 peptide of 7 amino acids long were accepted, and transfer of peptide identifications between runs was enabled. The peptide-spectrum match and protein FDR were kept below 1% using a target-decoy approach with reversed sequences as decoys.

In heavy-spiked samples, normalized H/L ratios (by shifting the median peptide log H/L ratio to zero) were used in all downstream quantitative analyses to account for any H/L 1:1 mixing deviations. Protein H/L values themselves were derived by using the median of a protein's peptide H/L ratios and required at least one peptide ratio measurement for reporting quantitative values. Signal integration of missing label channels was enabled. For enriched phosphoproteome samples, an in-house-written R script based on median phosphopeptide intensity was used to normalize the phosphopeptide intensities.

The heavy spike-in standard used for deriving the copy numbers was quantified using the iBAQ method as described by ref. [48]. Essentially, UPS2 protein intensities were divided by the number of theoretically observable peptides, log-transformed and plotted against log-transformed known protein amounts of the UPS2 proteins. This regression was then applied to derive all other protein absolute quantities using each protein's iBAQ intensity. The relative ratios of individual proteins to total protein were then converted to protein concentration in the cell by multiplying the total protein content in the cell for each condition. The total protein content per cell under each condition was measured using the modified Lowery method.

**Phosphorylation regulation analysis.** A recently developed method[33] for FPE identification was used for phosphorylation regulation analysis in this work. For details of the model and method, the readers are referred to the original publication. Briefly, it has been shown that the correlation between changes in fluxes and phosphorylation levels suggests the contribution of phosphorylation events to the fluxes. A phosphorylation event is inferred to activate enzyme activity if the correlation is positive while inhibiting enzyme activity if negative. Therefore, correlation analysis was performed in this study for the fold-change values of fluxes and phosphopeptide intensities by comparison with a reference dilution rate.

**Relative metabolome quantification[49].** For intracellular metabolomics analysis, frozen biomass pellets were delivered to Metabolon, Inc. (Durham, NC, USA), where nontargeted MS was performed. Briefly, metabolites were identified by matching their ion chromatographic retention index and MS fragmentation signatures to the Metabolon reference library of chemical standards. Relative quantification of metabolite concentrations was then performed via peak area integration.

**Metabolite quantification and data normalization.** Peaks were quantified using the area under the curve. For studies spanning multiple days, a data normalization step was performed to correct variation resulting from instrument interday tuning differences. Essentially, each compound was corrected in run-day blocks by registering the medians to equal one (1.00) and normalizing each data point proportionately (termed the "block correction"). For studies that did not require more than 1 day of analysis, no normalization was necessary, other than for purposes of data visualization.

**Bayesian inference details**
*Derivation of loglinear kinetics based on thermodynamics.* The linear relation between the reaction rate and reaction affinity proposed by ref. [50] is as follows:

$$v = eLA \tag{2}$$

where $L$ is the phenomenological coefficient, and $A$ is the reaction affinity (which equals minus the change in free energy of the reaction). Here, we added the enzyme amount term ($e$) to the original equation, and the same expression form was also discussed in Visser[51]. It may be argued that the relation of Eq. (2) is only valid close to equilibrium; however, many empirical analyses have observed that the linear relationship between the reaction rate and reaction affinity is valid even if the reaction operates far from equilibrium[52–54].

The reaction affinity term was then substituted by the 2nd thermodynamic law equation as follows:

$$A = RT \cdot \ln\left(\frac{K_{eq}}{Q}\right) \tag{3}$$

where $K_{eq}$ is the reaction equilibrium constant, and $Q$ is the reaction quotient. Taking the following reaction as an example:

$$a\,A + b\,B \overset{e}{\longleftrightarrow} c\,C + d\,D$$

According to Eqs. (2) and (3), the net forward reaction rate can be expressed as follows:

$$v = eLRT \cdot \left(a \cdot \ln\frac{[A]}{[A^*]} + b \cdot \ln\frac{[B]}{[B^*]} - c \cdot \ln\frac{[C]}{[C^*]} - d \cdot \ln\frac{[D]}{[D^*]}\right)$$

$$v = e \cdot \left(LRTa \cdot \ln\frac{[A]}{[A^*]} + LRTb \cdot \ln\frac{[B]}{[B^*]} - LRTc \cdot \ln\frac{[C]}{[C^*]} - LRTd \cdot \ln\frac{[D]}{[D^*]}\right) \tag{4}$$

$$v = e \cdot \sum_{i=1}^{4}\left(a_i \cdot \ln\frac{[X_i]}{[X_i^*]}\right)$$

Under one steady state $c$, the reaction rate $v$ can be expressed in the form of flux $J$. Dividing the steady state flux by enzyme concentration will give an enzymatic specific flux $j$. Under the specific steady state condition $c$, it gives:

$$j^c = \sum_{i=1}^{4}\left(a_i \cdot \ln\frac{[X_i^c]}{[X_i^*]}\right) \tag{5}$$

It should be pointed out that $a_i$ is a coefficient independent of the steady-state conditions and corresponds to the allosteric effect of each metabolite. Furthermore, $a_i$ has the same dimension as $k_{cat}$ and the enzyme turnover number; thus, we call these coefficients the intrinsic turnover number ($k_{cat,intrinsic\_i}$, which is the intrinsic turnover number of enzymes for metabolite $i$) with respect to individual metabolites that take part in the reaction. With this concept, we can even include allosteric effectors in Eq. (5).

However, to apply Eq. (5) in integrating multiomics data, we also need to eliminate the equilibrium terms ($X^*$) in the equation. We then take a specific steady state as the reference state denoted by superscript 0, and the kinetics equation under the reference state is as follows:

$$j^0 = \sum_{i=1}^{4}\left(a_i \cdot \ln\frac{[X_i^0]}{[X_i^*]}\right) \tag{6}$$

By subtracting Eq. (6) from Eq. (5), we obtained the final model we used to integrate fluxome, proteome and metabolome data. It can be expressed as follows:

$$\frac{J_{pred}^c}{e^c} = \frac{J^0}{e^0} + \sum_{i=1}^{4}\left(a_i \cdot \ln\frac{[X_i^c]}{[X_i^0]}\right) \tag{7}$$

The above kinetic equation separates the effect of enzymes and metabolites on the reaction flux by using enzymatic specific flux instead of the reaction flux itself. It can be derived that the difference in enzymatic specific flux under the two conditions is determined by the relative change in metabolite concentration and their corresponding kinetics parameters (intrinsic turnover number, $a_i$).

*MCMC Bayesian inference of the intrinsic turnover number for each metabolite.* Given a reference state (with enzyme abundance ($e^0$), fluxes ($J^0$)), the linear thermokinetic equation (Eq. (7)) translates absolute enzyme abundance ($e^c$), relative metabolite abundances with respect to reference ($X^c/X^0$) and intrinsic turnover numbers ($k_{cat,intrinsic\_i}$, i.e., $a_i$ in Eq. (7)) into the predicted flux ($J_{pred}^c$) under Condition $c$. To determine whether an investigated reaction obeys the above thermokinetic equation (Eq. (7)), we must find a set of kinetics parameters that best fit the FBA-determined flux ($J_{obs}^c$). As there have been no reports on the values of the proposed intrinsic turnover numbers, we want both a maximum a posterior probability estimator of $a_i$ and a measure of parameter uncertainty. To do this, we applied a Bayesian inference approach to estimate these kinetic parameters $a_i$:

$$\Pr(a_i, |, J_{obs}, X, e) = \frac{\Pr(J_{obs}, X, e, |, a_i)\Pr(a_i)}{\Pr(J_{obs}, X, e)}$$
$$\Pr(a_i, |, J_{obs}, X, e) \propto \Pr(J_{obs}, X, e, |, a_i)\Pr(a_i) \tag{8}$$

The posterior distribution of parameter ($a_i$) was estimated using Markov chain Monte–Carlo-based Bayesian inference, and the open source Python Bayes package PyMC3[55] was used. A prior distribution of the model parameter $\Pr(a_i)$ was first proposed, and samples of $a_i$ were drawn from the prior distribution. Then, the reaction flux $J_{sim}$ was evaluated using these drawn $a_i$ values through Eq. (7), following which a log-likelihood of $a_i$ ($\Pr(J_{obs}, X, e, |, a_i)$) is used to determine how well the predicted flux agrees with the flux data ($J_{obs}$). Finally, the posterior probability of these drawn $a_i$ were calculated using Eq. (8), and it was determined whether the drawn $a_i$ should be accepted or rejected. Iteratively, the above steps of drawing the parameter from the prior, evaluating the predicted flux, calculating the log-likelihood and finally determining whether to keep or reject the drawn $a_i$ sample were repeated until the upper bound of the iteration steps. All except $a_i$ form the posterior distribution space, which will reveal the most likely value for each $a_i$ and the credibility interval.

Without any experience of the prior distribution of $a_i$, we assume a normal distribution for these parameters

$$a_i = \text{Normal}\left(\mu = \text{mean}(j_{obs}), \delta^2 = \{\max(\text{std}(j_{obs})^2)\}\right)$$, i.e., with the mean equal to be the mean of the observed turnover number and variance to be the square of the standard error of the turnover number. FBA analysis combined with FVA was carried out using Yeast-GEM v 7.6, and both point estimation and variation of fluxes were obtained. Point estimation of the flux was used to calculate the squared error $\delta_c$ between the observed and predicted fluxes with Eq. (8) using $a_i$ drawn from the prior distribution. Similar to Hackett[15], we assume that the deviation between $j_{obs}$ and $j_{pred}$ followed a normal distribution with variance given by the squared error. With the help of FVA, the experimental uncertainty was introduced through flux variability. The log-likelihood of $a_i$ was then modified to account for the flux variability for the estimation of the posterior distribution of $a_i$. The modified log-likelihood function is as follows:

$$l(a_i, |, J_{obs}, X, e) = \sum_{c=1}^{n} \log\left[\frac{\varnothing_{\mu=\bar{j}_{pred}^c, \delta^2=\delta_c^2}(j_{obs}^{c,upper}) - \varnothing_{\mu=\bar{j}_{pred}^c, \delta^2=\delta_c^2}(j_{obs}^{c,lower})}{j_{obs}^{c,upper} - j_{obs}^{c,lower}}\right] \tag{9}$$

$\varnothing$ is the cumulative distribution function of the normal distribution with parameters $\mu$ and $\delta^2$. $\bar{j}_{pred}^c$ denotes the predicted specific flux under condition c, and $j_{obs}^{c,upper}$ and $j_{obs}^{c,lower}$ are the upper and lower values of flux estimated using FVA.

$\delta_c^2 = \frac{\sum_{c=1}^{n}\left(\bar{j}_{pred}^c - j_{obs}^c\right)^2}{n-I}$, where $n$ denotes the number of experimental conditions and $I$ denotes the number of metabolites involved in the model.

The log-likelihood function combined with the prior distribution of model parameters was then used to calculate the posterior probability of the drawn $a_i$. Using the MCMC algorithm, the decision of dropping or keeping the drawn $a_i$ was made. Large numbers of iterative steps (20,000 samples with 1000 draws discarded between two consecutive samples, so 120,000 steps in total) were needed to guarantee convergence, and the $\hat{R}$ value calculated by the Gelman-Rubin statistic method[56] was used as the criterion to confirm convergence of the inferred parameters.

**GECKO modeling details.** The enzyme-constrained model ecYeast7, version 1.4, was used from release 1.1.1 of GECKO: https://github.com/SysBioChalmers/GECKO/releases/tag/v1.1.1. A fraction of metabolic enzymes of $f = 0.4461$ g/g was

assumed based on Pax-DB data, and an average saturation of $\sigma = 0.49$ was assumed for any nonmeasured enzyme. The measured protein content was rescaled to be proportional to previous measurements of 0.46 g/gDW at $0.1\ h^{-1}$. Manual curation was performed to some $k_{cat}$ values, exchange fluxes of pyruvate, acetaldehyde and (R, R)−2,3-butanediol were blocked, and a non-growth-associated maintenance of 0.7 was assumed for all growth conditions.

For each condition, the corresponding rescaled proteomic data were overlaid as constraints on any protein that had a match, removing zero values. An additional standard deviation was added for each protein to prevent overconstrained models, and all four complexes from oxidative phosphorylation were scaled to be proportional to the average measured subunit. For all undetected enzymes, an overall "pool" constraint was used, equal to the difference between the protein content and the sum of all measured proteins, multiplied by the previously mentioned $f$ and $\sigma$. Additional details plus the full implementation of this process are available in the script *limitModel.m*.

For each condition, the previously obtained model was used to fit the chemostat data by modifying the growth-associated maintenance and optimizing for biomass growth. The implementation of this is available in *dilutionStudy.m*. Finally, as several proteins were shown to be a large limitation due to an extremely low detected measurement, the mentioned models were flexibilized so they could at least grow at the desired biomass growth rate with the available glucose. This implementation is available in *flexibilizeModels.m*.

**Total protein content measurement**. The total protein contents of each condition were measured using the modified Lowry method[57]. The total protein contents measured at all dilution rates ($n = 3$, ±standard deviation) are shown in Supplementary Table 6.

**Reporting summary**. Further information on research design is available in the Nature Research Reporting Summary linked to this article.

## Data availability
RNA-seq raw data that support the findings of this study have been deposited in NCBI's SRA with the accession code: PRJNA523289. Proteome and phosphoproteome raw data that support the findings of this study have been deposited in ProteomeXchange's PRIDE Archive with project accession code: PXD012891. Metabolomics data were generated by METABOLON. The processed data are available on github at https://github.com/bioexplore/multiomcispaperdata/tree/main/MetabolomeData. The mass spectrometry raw data related have been deposited to MetaboLights with study number MTBLS697. The genome sequence of CEN.PK113-7D is available at https://www.yeastgenome.org/strain/S000203459. Source data are provided with this paper.

## Code availability
GECKO v1.1.1 was used for enzyme usage analysis and is available at https://github.com/SysBioChalmers/GECKO/releases/tag/v1.1.1 Other custom codes or scripts are provided as Supplementary Software, including detailed descriptions of all codes in the accompanying readme files.

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

## Acknowledgements

The authors would like to acknowledge the Project 21776082 (J.X. and J.N.) supported by National Natural Science Foundation of China, the Knut and Alice Wallenberg Foundation, the Novo Nordisk Foundation (grant no. NNF10CC1016517, J.N.), the Swedish Foundation for Strategic Research and DD-DeCaF (Horizon2020 project 686070, J.N.). We would like to thank Anna Koza for conducting the RNA-Seq measurement of our samples, Petri-Jaan Lahtvee for his help on the absolute transcriptome and proteome measurement method discussion, Yongjun Wei for his help on the experiment conducting and large amount of omics samples collection.

## Author contributions

J.X. conceived the experiment, did the omics data integration analysis and Bayesian inference modeling and draft the manuscript. B.J.S. did the genome-scale network simulation work with proteome constraints using GECKO and help to edit the manuscript. Y.C. did the phosphorylation regulation analysis part phosphoproteome analysis and help to edit the manuscript. K.C. did help the multiomics sample processing and draft and revise the manuscript. S.K. did the proteome and phosphoproteome measurement. J.N. designed the experiment and support all the omics data measurement and discuss for forming the manuscript and read and revised the manuscript.

## Funding

## Competing interests

The authors declare no competing interests.
