## [Peer Review File · Nature Communications]

REVIEWER COMMENTS

Reviewer #1 (Remarks to the Author):

May 29th, 2021

Review of Xia et al, "Proteome allocations change linearly with specific growth rate of *Saccharomyces cerevisiae* under glucose-limitation".

In this study, the authors use *S. cerevisiae* growing in a very broad range of glucose availability, including in respiratory, respire-fermentative, and fermentative conditions, to assess the reallocations of proteome, mRNA and metabolome. This follows excellent recent studies from the same group on translational reserves after nitrogen starvation (Yu et al, 2020), and proteome allocations towards the ribosome (Björkeröth J et al 2020). They propose that several categories of proteins and mRNA change in a function specific manner with increasing growth rate. An emphasis is on glycolytic and mitochondrial proteins (as a proxy for respiro-fermentative status), translation proteins, chaperons and lipid biosynthesis proteins (as a proxy for growth status). Using absolute transcriptomics data, they make a case that the variability in protein levels correlate well to mRNA. One mechanistic explanation proposed for their observed results is a claim that decreased phosphorylation increases enzymatic activity in a non-specific manner (particularly for glycolysis). According to the authors, this leads to higher flux through the pathway against reduced enzyme levels.

Overall, this is an interesting and well designed/executed study, and has substantial fantastic multi-omic data obtained for the first time. However, several parts of the manuscript suffer from over interpretations, not fully supported by data, and also some points stated do not acknowledge prior bodies of work. Some of the correlations are used to over-claim the mechanisms underlying the observations. Particularly, one of the major conclusions of the study, i.e. phosphorylation based control of enzyme activity entirely lacks experimental validation or prior evidence, but makes a strong statement of certainty ignoring possible confounding observations. Also, in many parts of the manuscript, the authors have used R² (Coefficient of determination) to state correlation. R² without directionality tests can easily confound interpretations (or lead to over interpretations). The correlations between two variables must be measured by Pearson (for normally distributed), or Spearman (for non-normally distributed) data. This manuscript requires a serious, careful relook (and in parts analysis), some reassessments of statements/conclusions, and a careful assessment of statements made, but the datasets themselves are fantastic.

Main comments:

1. The authors used coefficient of determination (R^2) to determine the correlation between two variables (especially note Figure 2A). While they mention in Figure 2B, that they have calculated the Pearson correlation coefficient, from Figure 2A it looks like this is the R^2 . R^2 only estimates how well the linear regression fits, but does not provide any direction of correlation. You can get a high R^2 for a weak but consistent correlation. To properly measure the strength and direction of the correlation, typically a Spearman correlation (For non-normally distributed data) will be used. The authors seem to have not noticed that the mRNA molecules per cell and protein molecules per cell are not normally distributed.

2. Did the authors carefully check the distribution before choosing to present Figure 2A? Relatedly, for the highlighted proteins in Figure 2A, do they all have established biological roles to growth rate variations?

3. What are the p-values and spearman rank correlations for the individual correlation coefficients corresponding to each mRNA-protein pair? Were they corrected for multiple hypotheses? It is not clear what FDR values were used, and how this was selected.

From Fig 2 (mainly 2A and 2B), you can only strongly conclude that proteins change with respect to growth rate, unless more systematically, properly analyzed.

Also, related to Figure 2:

4. There are no error bars for Figure 2C, 2D. If this was a weak weak correlation, it could well be insignificant (eg. for amino acid biosynthesis, slope of 0.09).

5. The authors claim that the mRNA level determines the protein levels in different growth conditions. If you compare Figure 1C and 1D with Figure 2C and 2D, what is clear is that protein levels are strongly correlated (based on slope) with growth rate. Not so with mRNA. For example, a slope of 0.18 for protein fraction for amino acid biosynthesis proteins vs 0.09 for mRNA fraction. How do the authors explain this? Again, this can come if the coefficient of determination is used instead of a coefficient of correlation.

6. Relatedly, Line 198-201 related to dynamic protein allocation being determined by mRNA level: can the authors use these data to define mRNA subsets, which can then predict growth rates accurately (across a broad range of carbon availability?). If so, this would be a predictive statement.

7. In Figures 1C, 1D, 2C and 2D, the correlation would depend on how the proteins are clustered and how many proteins are chosen for a category. Here, the authors have used a classification adopted by a previous study by MetzI-Raz et al.

a) As an essential control, the authors should conduct a leave-one-out analysis of proteins in each functional cluster to check if that strengthens or weakens the correlation between the protein fraction in for a given category and the growth rate. For example, a set of proteins in a given category will have negative correlations while another set will have positive correlations. The observed correlation indicates lumped effects of all the proteins. This does not provide much biological insight.

b) Like with individual mRNAs, the authors should similarly conduct a correlation analysis at individual protein levels with specific growth rates.

8. Related to the statement “Two other groups (amino acids biosynthesis and mitochondrial proteins) also showed increasing tendency along with the specific growth rate (Fig.1C). However, unlike the fraction of the translation proteins, these two groups had a profile that was metabolism depending (Fig.1C).”

It will be utterly surprising if the results were anything else, since both are so tightly part of metabolism, and amino acid biosynthesis is itself heavily dependent on mitochondria. A substantial proportion of all amino acid biosynthesis enzymes are mitochondrial. Hence, the amount of amino acid biosynthesis enzymes will correlate with mitochondrial protein absolute amounts, and mitochondrial protein amounts are well established to change depending on fermentative, respiratory or respiratory-fermentative conditions. So what is the point the authors are trying to make, apart from making the well established sound like something novel? A simpler, more useful analysis will be to see how well amino acid biosynthesis transcripts and proteins correlated with mitochondrial ones, and make a simpler, minor point.

9. Relatedly, for the statement/results on lipid synthesis being “at a constant level during respiratory metabolism, but this fraction decreased when the cells shifted to respiro-fermentative metabolism.”

Here too, the authors need to more carefully consider what is known. A large part of lipid biosynthesis (especially phospholipids and cardiolipin) are all entirely mitochondrial. Hence, some decrease as cells shift to respire-fermentative metabolism will be expected. Please clarify what is the point you are trying to make.

10. Line 186: More than mRNA fractions in lipid, glycolysis etc being reversely proportional to growth rates (data ambiguous, statistics poor), it seems that these are positively correlated with specific metabolic states (respirative, respiro-fermentative, and fermentative). Please analyze both ways.

11. Related to figure 3: the theoretical metabolic flux was estimated using GECKO and the genome-scale metabolic model of yeast. They correlate the flux with mRNA and protein levels separately, and look for reactions which show positive or negative correlation for both mRNA and protein. Some of this is quite contrived:

- In Line 216, they mention that TCA cycle & nucleotide biosynthesis show strong positive correlation for both mRNA and proteins. I cannot find a proper functional enrichment analysis for these, with significantly affected metabolic modules. Were by just looking at the metabolic map obtained from iPath3? While this might be so, it needs to be tested properly.

- For Figure 3C, the authors can show simple bar graphs and present the interconnection between the multi-omics data. Currently, it's not readable at all.

12. How strong is the correlation between phosphorylation status of the enzyme and the metabolic flux? This is the weakest section of this manuscript. There are some mechanistic studies that look at phosphorylation of multiple metabolic enzymes, but no literature is cited that shows that increased phosphorylation leads to enzyme inhibition. Is there sufficient/any experimental evidence on this? If not, it is extremely premature for the authors to conclude this as a mechanism of how high flux is maintained despite enzyme concentrations.

13. The authors do not sufficiently explore the more obvious: the effect of metabolite concentrations in flux even at decreased enzyme concentrations, and how that responds to increasing growth rates. Further related to Fig 3C etc, on the specific activity of glycolytic enzymes or their usage, the authors systematically clarify then on how much of a 'buffer' of these enzymes are present? How much then is determined by mass action/substrate availability itself? Multiple studies (notably in yeast Hackett et al PMID: 27789812) quite clearly state that substrate concentrations are the primary drivers of metabolic reactions). So, in many ways the data in this paper primarily substantiates these findings systematically. There is no need to present this as novel, earth-shattering information.

14. In Fig 4D again the authors have presented R². Not clear if the coefficient of correlation or the coefficient of determination for linear regression between amino acid biosynthesis intermediates and the specific growth rate is shown.

15. Also related to fig 4B and D: related lysine and arginine biosynthetic pathway genes with a positive correlation with growth rate. Can the authors see what the correlation of these pathway enzymes and transcripts, with ribosome biogenesis, across these broad growth rates/carbon availability? Recent experimental studies show that the synthesis and availability of lysine and arginine are the primary determinants of ribosome biogenesis (PMID: 33378328), apart from more extensive literature showing that the ribosomal proteins are extremely arginine and lysine rich. Of course, there is the very well established correlation of ribosome biogenesis with growth itself (from

the Warner et al papers to more recent studies). So this seems like a much more plausible, biologically relevant correlation.

Also related to Fig 4 (lines 357-62 etc): "Increased enzyme concentration and enzyme saturation in amino acid metabolism further supports this flux increase for increasing growth rates." This is fully consistent with Hackett et al 2004 and other studies. Also see your lines 395-398, which states the same point (without including these references).

More minor points:

1. Line 351, in order to explain the cause behind low amino acid concentration despite high tRNA synthetase abundance, one needs to calculate the flux through the tRNA synthetase (which can be easily done using the metabolic model) and check if that correlates with the tRNA synthetase abundance.

2. There is some slightly careless referencing. For example, refs 6 and 7 are hardly illustrative of the use of yeast as "a model organism for deciphering molecular mechanisms in cellular and molecular biology", but this made me smile. Please go through the references carefully, and appropriately cite past literature.

3. Line 139: There is no strong evidence (or backing references, in lieu of evidence) with which to state that decrease in glycolytic and chaperon fraction is strategically important for making space for other proteins.

3. Figure 4A and 4C are noisy and very difficult to read. Can the authors present only the enzymes which show significant changes with respect to the independent variable? The rest can be a supplemental

4. Several typos in the manuscript. Please proof carefully. For example, correct spelling of slope (slop in places), line 227 “....important role to fulfill...” and many, many more

Reviewer #2 (Remarks to the Author):

Xia et al. describe a study in which they cultured yeast in chemostats with glucose as the sole carbon source. They used 9 different dilution rates that equal the specific growth rate (ranging from 0.025 per h to 0.4 per h) and measured the transcriptome, proteome, phosphoproteome and metabolome. Additionally, the authors inferred fluxes from measured exchange fluxes of glucose, ethanol, O₂ and CO₂. Main findings are: i) the concentration of many proteins (or groups of proteins like ribosomal proteins) is growth rate dependent, ii) phosphorylation of glycolytic enzymes and chaperons is growth rate dependent and iii) proteome allocation is determined at the transcriptome level. The strength of the work is the well-designed experiment that employs chemostats to enforce different growth rates and different metabolic states (respiratory vs respiro-fermentative metabolism) in yeast. The data set is impressive, of high quality and probably one of the most comprehensive multi-omics data sets. My main concern is that the analysis of the data is very descriptive and focuses mostly on correlations between growth and the different “omics” levels. Specific point are:

1) A large number of studies examined growth-rate dependencies of transcription, translation and metabolism and the authors do a great job explaining them in their introduction. However, the manuscript should better describe which growth-rate dependencies have been observed by others (and in which organism) and which growth rate effects are newly revealed by their data. For example, MetzI-Raz et al (Ref [8] in the text) observed similar relationships in yeast, but it is unclear if there are differences to the data shown here and if they are due to the different conditions.

2) In Figure 1 C and D the authors fit two linear functions to one proteome category. What is the basis to assign data points to the first or the second function (e.g. in Fig 1C there are 3 data points in the second fit and in Figure 1D there are 4 data points in the second fit)? From Figure 1 B it seems that only the two highest growth rates are in the respiro-fermentative regime.

3) The data should contain information about the regulatory mechanisms that lead to the changes in the proteome fraction. Since the authors nicely show that the changes occur at the transcriptional level, they could go one step further and uncover the transcriptional regulators that are involved.

For example, which regulator could be responsible for downregulating glycolysis enzymes? At least the authors can discuss some hypothesis about this.

4) It is unclear if the authors considered growth-rate dependency of cell volume. For many organisms the cell volume changes with the growth rate. If such cell volume changes also occur in yeast they can substantially impact the results.

5) The authors could use a more systematic approach to integrate the different omics levels in order to extract the full information from their data set. The insights from the current analysis reflect to a large part the findings in *E. coli* (Scott et al., Reference [11] in the text) and yeast (Hackett et al., [13]). For example, in line 280 hierarchical regulation analysis is mentioned, which is indeed a great tool to integrate all data types and test how the different levels contribute to flux regulation. But the analysis was restricted to i) only the proteome and ii) only the central metabolism and amino acid synthesis. Did the authors attempt to estimate the coefficients of all enzymes, metabolites and PTMs (as they mention themselves in line 281)?

6) In the last part of the model it is not clear how and why enzyme usage was calculated with the GECKO method. Enzyme usage can be directly estimated for each enzyme individually using fluxes, k_{cat} and the proteome data. The authors should discuss the advantage/difference of the GECKO method.

7) In the discussion the authors claim they “mapped the multilayer regulation structure in protein allocation”. I don’t agree with this statement because the authors only show correlations between the different layers but don’t infer regulatory interactions or show which mechanisms are active. For example, the decreasing concentrations of amino acids is an interesting finding (line 359), but the authors cannot show if they are affecting the respective feedback mechanism and at which layer (enzyme activity or enzyme concentration).

Reviewer #3 (Remarks to the Author):

Overview

Xia and coauthors provide a detailed multiomic characterization of *S. cerevisiae* growth across a range of specific growth rates in carbon-limited chemostats. Faster growth requires cells to make a similar amount of biomass in less time, and also encompasses the well known Crabtree effect. As a result, understanding the regulatory processes operating across this regime is of considerable interest. The experimental methodology and informatics used by the authors appears to be sound, and they report three major findings. First, separating proteins and transcripts into major cellular processes suggests that cells' investment in these different categories is similar at the transcriptional and protein level. Thus, it is arguable that gross regulation of protein abundance is primarily at the transcriptional level. Second, increased activity of chaperones and glycolytic enzymes is consistent with a decrease in phosphorylation with specific growth rate. Third, increased amino acid biosynthesis can be partially explained by increased enzyme saturation. Of these findings, the regulation of glycolytic flux by dephosphorylation is arguably the most important result but I have my doubts about the validity of this finding since metabolic regulation was not considered as a competing hypothesis. Interpretation of the results is also made more difficult by relatively poor writing and unpolished figures.

Major

1. The author's assertion that glycolytic flux control occurs primarily through phosphorylation ignores the metabolomics data that they have collected. Boer et al. 2010 "Growth-Limiting Intracellular Metabolites in Yeast Growing under Diverse Nutrient Limitations." *Molecular Biology of the Cell* indicates that upper glycolytic intermediates increase with specific growth rate in carbon limited yeast chemostats. To wit, increases in substrate saturation likely contribute to growth-rate dependent elevation of glycolytic flux. The authors should examine the influence of hierarchical control versus metabolic control in glycolysis simultaneously as they did for amino acid synthesis.
2. Comparing "fraction of the proteome" to "fraction of the transcriptome" to make the argument that "proteome allocation pattern is mainly determined at the mRNA level" is difficult. To make this point the authors should directly compare these quantities rather than relying on readers comparing Figures 1 and 2. For example, if their assertion is true, the ratio of proteome fraction to transcript fraction should be independent of specific growth rate. Alternatively, a bivariate scatter plot could demonstrate proportionality.
3. The paper requires thorough proofreading. A subset of the word usage, formatting and major english errors can be found in minor.

Minor

1. In several instances the authors report counts of genome-scale measurement p-values less than a cutoff without applying false-discovery rate control. For example, “We filtered out significant (p value < 0.05)”.

2. To make the case that the levels of unphosphorylated glycolytic enzyme increases with specific growth rate more compelling, the authors should add this information to Figure 3C. The levels of the total enzyme and phospho enzyme look really similar, which would mean that the phosphorylation events could be totally inert.

3. The author’s overstate “Both hierarchical regulation analysis and FPE inferring confirmed function inactivation due to phosphorylation of seven of the glycolysis enzymes” as well as “We therefore conclude that phosphorylation of both glycolytic enzymes and chaperones seems likely to reduce their activities”. Based on my reading of FPE, it is primarily picking up on the anticorrelation of flux and phosphorylation/phospho-protein levels. Unless competing hypotheses can be ruled out (see major 2), its more appropriate to indicate that increased flux could be caused by [or, is consistent with] a decrease in inhibitory phosphorylation events.

4. The authors should justify why their analysis of protein phosphorylation occurs at the protein-level rather than site-level. Some modifications may affect catalysis while others would not. I can imagine the real reason here is that we don’t know the role of most phosphosites, and like in conventional proteomics, the phosphopeptides we obtain are just a sample of the total set. This is reasonable, but caveats should be spelled out.

5. Please provide additional details on Figure S8. It is unclear why the role of hierarchical control seems to fluctuate so much as a function of specific growth rate when most measures vary smoothly.

6. Describing whether the previously reported “phosphorylation events on Pfk2, Fba1 and Gpm1” were also found to be inhibitory by Oliveira et al. would more strongly support the findings of this study.

7. Consider specifying that the slope from the RNAseq, absolute mRNA regression was used to estimate the absolute mRNA for the remaining genes. Mentioning the R2 statistic then that “The same correlation was applied” may be confusing.

Figure 1C doesn’t look like an R2 of 1, how many significant digits are used?

8. In 1C/D, "proportionality" is inappropriate; in these instances correlation would be better.

9. To make Figure 3 cleaner the authors should consider increasing the resolution of the legend text in 3A/B and removing unnecessary annotations on reaction map (3B) and axis text (3C).

10. "majority of the dry cell weight" implies > 50% of the total, while protein ranges from 30-50%.

11. English, grammar: "P values were gotten from", "two groups increased indeed linearly", "these two groups had a profile that was metabolism depending", "abundance of proteins are 3 orders bigger", "the functional parts of the life process", "Enzyme saturation play an important role fulfil amino acid biosynthesis", "We are confident that these findings will besides providing abasic insight into regulation of metabolism in eukaryal model organism *S. cerevisiae* it will also assist..."

12. Formatting, missing characters: "-20 until"

13. Formatting, typos: "fulfil", "slop", "underlyingprinciples", "2302fluxes", "Mitochodira", "Chaperons" (Figure S5)

14. Formatting, excess whitespace: "volume before"

REVIEWER COMMENTS and OUR RESPONSES

Reviewer #1 (Remarks to the Author):

May 29th, 2021

Review of Xia et al, “Proteome allocations change linearly with specific growth rate of *Saccharomyces cerevisiae* under glucose-limitation”.

In this study, the authors use *S. cerevisiae* growing in a very broad range of glucose availability, including in respiratory, respire-fermentative, and fermentative conditions, to assess the reallocations of proteome, mRNA and metabolome. This follows excellent recent studies from the same group on translational reserves after nitrogen starvation (Yu et al, 2020), and proteome allocations towards the ribosome (Björkeröth J et al 2020). They propose that several categories of proteins and mRNA change in a function specific manner with increasing growth rate. An emphasis is on glycolytic and mitochondrial proteins (as a proxy for respiro-fermentative status), translation proteins, chaperons and lipid biosynthesis proteins (as a proxy for growth status). Using absolute transcriptomic data, they make a case that the variability in protein levels correlate well to mRNA. One mechanistic explanation proposed for their observed results is a claim that decreased phosphorylation increases enzymatic activity in a non-specific manner (particularly for glycolysis). According to the authors, this leads to higher flux through the pathway against reduced enzyme levels.

Overall, this is an interesting and well-designed/executed study, and has substantial fantastic multi-omic data obtained for the first time. However,

1. Several parts of the manuscript suffer from over interpretations, not fully supported by data, and also some points stated do not acknowledge prior bodies of work. Some of the correlations are used to over-claim the mechanisms underlying the observations.
2. Particularly, one of the major conclusions of the study, i.e. phosphorylation based control of enzyme activity entirely lacks experimental validation or prior evidence, but makes a strong statement of certainty ignoring possible confounding observations.
3. Also, in many parts of the manuscript, the authors have used R² (Coefficient of determination) to state correlation. R² without directionality tests can easily confound interpretations (or lead to over interpretations). The correlations between two variables must be measured by Pearson (for normally distributed), or Spearman (for non-normally distributed) data.
4. This manuscript requires a serious, careful relook (and in parts analysis), some reassessments of statements/conclusions, and a careful assessment of statements made, but the datasets themselves are fantastic.

Response: We thank the reviewer for the positive comments on our paper and will address these key points below.

Main comments:

1. The authors used coefficient of determination (R²) to determine the correlation between two

variables (especially note Figure 2A). While they mention in Figure 2B, that they have calculated the Pearson correlation coefficient, from Figure 2A it looks like this is the R2. R2 only estimates how well the linear regression fits, but does not provide any direction of correlation. You can get a high R2 for a weak but consistent correlation. To properly measure the strength and direction of the correlation, typically a Spearman correlation (For non-normally distributed data) will be used. The authors seem to have not noticed that the mRNA molecules per cell and protein molecules per cell are not normally distributed.

Response: Thanks for pointing out the difference among coefficient of determination, Pearson correlation coefficient, and Spearman correlation coefficient. We applied the Shapiro-Wilk normality distribution tests for both mRNA molecules per cell and protein molecules per cell, the results confirmed the non-normally distribution for both mRNA and protein molecules per cell data. Therefore, we revised our analysis using the Spearman correlation. Figure 2A and Figure 2B are both updated with the Spearman correlation coefficient to replace coefficient of determination and Pearson correlation coefficient, respectively. Python scripts for checking the normal distribution of mRNA and protein abundance together with the results are uploaded to github as supplementary file (https://github.com/bioexplore/multiomcispaperdata/tree/main/ResponsesToReviewers/Reviewer1/Q1_Prteome_and_Transcriptome_normality_check), which include:

1. “protein_mRNA_normality_check.ipynb”, jupyter notebook script
2. “proteome_data_normality_check.xlsx”, Shapiro-Wilk check results for proteome
3. “Transcriptome_data_normality_check.xlsx”, Shapiro-Wilk check results for transcriptome

2. Did the authors carefully check the distribution before choosing to present Figure 2A? Relatedly, for the highlighted proteins in Figure 2A, do they all have established biological roles to growth rate variations?

Response: Figure 2A is updated according to the previous comments. The distribution of the mRNA and protein were checked to be non-normally distributed, so the Spearman correlation coefficient was used instead. The highlighted proteins in Figure 2A were updated, and the updated proteins were chosen as follows:

1. We did Spearman correlation analysis between individual protein and corresponding mRNA, with $FDR < 0.05$. We got 54 protein-mRNA pairs with perfect correlation (correlation coefficient = 1 and p-value = 0).
2. For these 54 genes, we did GO term enrichment analysis. While no enrichment was found for GO Process term with $FDR = 0.05$, four enrichments were found for GO Function term: “phospholipid-hydroperoxide glutathione peroxidase activity” with 2 genes, “peroxidase activity” with 3 genes, “oxidoreductase activity” with 10 genes and “oxidoreductase activity, acting on peroxide as acceptor” with 3 genes.
3. There are 3 genes fall commonly in three of the four above GO Function terms and this was used to update Fig. 2A. The genes are *GPXI*, *CTA1*, and *HYR1*.

3. What are the p-values and spearman rank correlations for the individual correlation coefficients corresponding to each mRNA-protein pair? Were they corrected for multiple hypotheses? It is not clear what FDR values were used, and how this was selected.

Response: We did the Spearman correlation analysis for each mRNA-protein pair, and used FDR

< 0.05 to correct the p-values for multiple testing hypotheses. It was found that our former statement that mRNA level determines corresponding protein level is not fully supported after FDR correction. Around 40% of mRNA-protein pairs showed significant correlation and among them most were positive correlations, which means that more than half of the proteins' levels showed no statistical correlations to their corresponding mRNA levels. From previous studies on the relationship between mRNA level and corresponding protein level we could also not find any consistent conclusions, but some research (Koussounadis et al., 2015, Sci. Rep., Maier et al., 2009, FEBS Lett.) reported that a low correlation between protein and mRNA was observed while others (Zur and Tuller, 2012, Embo. Rep.) reported high correlations. Scripts for doing the Spearman correlation analysis and results are uploaded to github as supplementary files:

[https://github.com/bioexplore/multiomcispaperdata/tree/main/ResponsesToReviewers/Reviewer1/Q3 correlation analysis between prteins and mRNAs](https://github.com/bioexplore/multiomcispaperdata/tree/main/ResponsesToReviewers/Reviewer1/Q3%20correlation%20analysis%20between%20prteins%20and%20mRNAs)

Reference:

Koussounadis A, Langdon SP, Um IH, Harrison DJ, Smith VA. Relationship between differentially expressed mRNA and mRNA-protein correlations in a xenograft model system. *Sci Rep* 5, (2015)

Maier T, Guell M, Serrano L. Correlation of mRNA and protein in complex biological samples. *FEBS Lett* 583, 3966-3973 (2009)

Zur H. and Tuller T. Strong association between mRNA folding strength and protein abundance in *S. cerevisiae*, *EMBO Rep.* 13(3):272-277 (2012)

From Fig 2 (mainly 2A and 2B), you can only strongly conclude that proteins change with respect to growth rate, unless more systematically, properly analyzed.

Response: In the previous version of Figure 2 (mainly 2A and 2B), we did not conclude that protein levels change with respect to growth rate, but we tried to interpret that protein levels change in response to changing mRNA levels. However, we agree that the original version of Figure 2 did not apply proper statistical analysis. So, we did a thorough analysis using our proteome and transcriptome data according to the reviewer's comments. After a more systematical and proper analysis using Spearman correlation analysis together with multiple testing FDR (<0.05 used) correction, we found that the protein level is partly (36% of all detected proteins showed significant positive correlation with mRNA level) determined by their mRNA level, which is shown in Fig. R1. Accordingly, we changed our conclusion "mRNA level determine its protein level by individual or in functional groups" to "Individually, no more than 40% mRNA level determine their coding protein level".

Fig. R1. Histogram plot of Spearman correlation coefficient for each protein-mRNA pair across a wide range of specific growth rates. (This figure is also used as the updated version of Fig2B)

Also, related to Figure 2:

4. There are no error bars for Figure 2C, 2D. If this was a weak correlation, it could well be insignificant (e.g. for amino acid biosynthesis, slope of 0.09).

Response: There is no error bars for Figure 2C and 2D, this is due to the absolute mRNA quantification method we used. What we did to obtain the absolute mRNA quantification is as follows:

1. We first get FPKM values of 18 carefully selected mRNAs, which cover 80% of the dynamic range of mRNA expression at the reference condition ($D = 0.1 \text{ h}^{-1}$). Absolute concentrations for these 18 mRNAs were then measured using QuantiGene assay (Affymetrix, Santa Clara, CA, United States), so that a standard curve between FPKM and absolute concentration was obtained. All these results were obtained in a previous publication of our lab in 2017 (Lahtvee et al., 2017, Cell Systems).
2. Here in this study, we did the same experiment with the same dilution rate ($D = 0.1 \text{ h}^{-1}$) together with other eight dilution rates (D from 0.025 h^{-1} to 0.4 h^{-1}). FPKM values for each dilution rate were obtain using TopHat2, and according to statistics method used in TopHat2 to get statistical meaningful FPKM value for each experiment condition at least two replicates are needed (here we use triplicate samples for each dilution rate), so only one FPKM value was obtained for each dilution rate condition. The standard curve established by Lahtvee 2017 was utilized to calculate the absolute mRNA concentration of the same dilution rate ($D = 0.1 \text{ h}^{-1}$). And fold changes of individual genes between other dilution rate samples and the reference condition ($D = 0.1 \text{ h}^{-1}$) were obtained using DESeq2, which also used triplicate samples to obtain statistically solid results of fold change, so each condition had obtained only one absolute mRNA concentration value.
3. Based on this method, each dilution rate condition has only one absolute mRNA concentration determined, so we can therefore no provide error bars for Fig. 2C and 2D.

Reference:

Lahtvee P-J, et al. Absolute Quantification of Protein and mRNA Abundances Demonstrate Variability in Gene-Specific Translation Efficiency in Yeast. Cell Systems 4, 495-504.e495 (2017).

5. The authors claim that the mRNA level determines the protein levels in different growth conditions. If you compare Figure 1C and 1D with Figure 2C and 2D, what is clear is that protein levels are strongly correlated (based on slope) with growth rate. Not so with mRNA. For example, a slope of 0.18 for protein fraction for amino acid biosynthesis proteins vs 0.09 for mRNA fraction. How do the authors explain this? Again, this can come if the coefficient of determination is used instead of a coefficient of correlation.

Response: After a thorough reanalysis based on the reviewer's comments, we changed our conclusion for this part: the mRNA level can partly determine the protein level across different growth rates. Regarding comparison of Figure 1C and 1D with Figure 2C and 2D, we can only say that total proteins in some groups are strongly correlated with growth rate but not each protein level is strongly correlated with growth rate. For the different slopes obtained for regression of proteins and mRNAs vs specific growth rate, we noticed that not only proteins engaged in amino acid biosynthesis but also other groups showed lower slopes for mRNA vs specific growth rate than that for protein vs specific growth rate. There are several possible reasons: 1) There are several orders of magnitude difference between absolute values of mRNA (on average within the order of 10^0 molecules/cell level) and protein (on average within the order of 10^4 molecules/cell level) for the same gene, which may cause lower slopes when doing regression with specific growth rate; 2) Less than 40% protein-mRNA pairs showed significantly positive correlation based on Spearman correlation analysis, even for these 40% the correlation obtained by Spearman may explain the slope difference of the regressed lines for protein and mRNA vs μ ; 3) The same changing trend with the specific growth rate does not mean that there will be the same slope, little increase in mRNA level may cause bigger increase in its coding protein level, at least at the individual group level the mRNA sum levels showed the same changing trend as their paired protein sum level in the same function group (Fig. S5 in the revised supplementary text).

6. Relatedly, Line 198-201 related to dynamic protein allocation being determined by mRNA level: can the authors use these data to define mRNA subsets, which can then predict growth rates accurately (across a broad range of carbon availability?). If so, this would be a predictive statement.

Response: Based on reanalysis of our data, we think that we over-claimed the conclusion in the previous manuscript version. We observed almost the same changing trend of protein and mRNA as respect to specific growth rate on the category level, however, our data currently do not strongly support that the protein allocation is determined by mRNA level. The corresponding text in the manuscript is updated accordingly.

7. In Figures 1C, 1D, 2C and 2D, the correlation would depend on how the proteins are clustered and how many proteins are chosen for a category. Here, the authors have used a classification adopted by a previous study by Metzl-Raz et al.

a) As an essential control, the authors should conduct a leave-one-out analysis of proteins in each functional cluster to check if that strengthens or weakens the correlation between the protein fraction in for a given category and the growth rate. For example, a set of proteins in a given category will have negative correlations while another set will have positive correlations. The observed correlation indicates lumped effects of all the proteins. This does not provide much

biological insight.

Response: Thanks for this excellent suggestion for the leave-one-out testing idea. We now did this for each category. We checked the Spearman correlation between part fraction sum (with one protein data leave out) and total fraction sum under all experiment conditions for each protein, and for most of the results (over 83%), the perfect correlation (coef = 1, and p-val = 0) was observed. If we use FDR < 0.05 to correct the p values, all correlations are statistically significant, which means that the leave-one-out does not influence the overall correlations between category protein fractions with specific growth rate. The leave-one-out test results are supplemented as a separate file “Leave_One_Out_test_result20210925.xlsx”. The following hist plot showed the leave-one-out correlation coefficient. Based on this result we are confident that no single protein has a big influence on the category fraction trend with specific growth rate.

Fig. R2 Leave-one-out testing for the correlation analysis

Python scripts and data files are uploaded to github also as supplementary files (https://github.com/bioexplore/multiomcispaperdata/tree/main/ResponsesToReviewers/Reviewer1/Q7_Leave-one-out-test), which include:

1. “Leave_one_out_test.py” the python script doing the leave one test
2. “eLIFE-ProteinCategoriesModify20200527.xlsx” categories used in this study
3. “ProteinAbsConcentration20210924.xlsx” absolute concentration of individual proteins
4. “Leave_One_Out_test_result20210925.xlsx” results file generated

b) Like with individual mRNAs, the authors should similarly conduct a correlation analysis at individual protein levels with specific growth rates.

Response: We did the correlation analysis between individual protein and specific growth rate, and we found that 44% of the proteins showed significant correlation with the specific growth rate and with half to half positive and negative correlations with $FDR < 0.05$. The spearman correlation distribution hist plot is shown below. Considering the clear linear relationships for the categories shown in Fig. 1C and 1D, it seems that the cell organizes the proteome allocation according to their functions but not individual proteins. Corresponding python script and data files are uploaded to [github \(https://github.com/bioexplore/multiomcispaperdata/tree/main/ResponsesToReviewers/Reviewer1/Q7b_correlation_between_individual_protein_and_specific_growth_rate\)](https://github.com/bioexplore/multiomcispaperdata/tree/main/ResponsesToReviewers/Reviewer1/Q7b_correlation_between_individual_protein_and_specific_growth_rate).

Fig. R3. Distribution of Spearman correlation coefficients for individual protein vs specific growth rate

8. Related to the statement “Two other groups (amino acids biosynthesis and mitochondrial proteins) also showed increasing tendency along with the specific growth rate (Fig.1C). However, unlike the fraction of the translation proteins, these two groups had a profile that was metabolism depending (Fig.1C).”

It will be utterly surprising if the results were anything else, since both are so tightly part of metabolism, and amino acid biosynthesis is itself heavily dependent on mitochondria. A substantial proportion of all amino acid biosynthesis enzymes are mitochondrial. Hence, the amount of amino acid biosynthesis enzymes will correlate with mitochondrial protein absolute amounts, and mitochondrial protein amounts are well established to change depending on fermentative, respiratory or respiratory-fermentative conditions. So what is the point the authors are trying to make, apart from making the well-established sound like something novel? A simpler,

more useful analysis will be to see how well amino acid biosynthesis transcripts and proteins correlated with mitochondrial ones, and make a simpler, minor point.

Response: Yes, we agree that most amino acid biosynthesis enzymes are mitochondrial. But here what we want to emphasize is not that these two groups showed a similar changing trend, we just want to express that even though these two groups increase with specific growth rate this is not true for respiratory-fermentative condition, and this point is different with that for the translation protein group. We did not mean to establish that this is something novel and have revised the text accordingly.

9. Relatedly, for the statement/results on lipid synthesis being “at a constant level during respiratory metabolism, but this fraction decreased when the cells shifted to respiro-fermentative metabolism.”

Here too, the authors need to more carefully consider what is known. A large part of lipid biosynthesis (especially phospholipids and cardiolipin) are all entirely mitochondrial. Hence, some decrease as cells shift to respire-fermentative metabolism will be expected. Please clarify what is the point you are trying to make.

Response: It is correct that many proteins associated with lipid metabolism are located in the mitochondria. According to “Henry, Kohlwein and Carman, Metabolism and regulation of glycerolipids in the yeast *Saccharomyces cerevisiae*, Genetics, 2012, 190(2):317-349.”, lipid metabolism related enzymes locate across several cellular compartments: ER, lipid droplets, mitochondria, vacuole, cytoplasm, and plasma membrane, vacuole membrane and with several phospholipid synthesis regulatory proteins located in nucleus, ER membrane etc. (Detailed information can be found in Table 1 of Henry, 2012, Genetics). We also did a GO Term (Cellular Component) enrichment analysis with our lipid metabolism category proteins, which showed that the most abundant enrichment component is endoplasmic reticulum (61 among 123) while enrichment for mitochondrion is much less (41 among 123). Thus, it is only about one third of the lipid biosynthetic genes that are associated with the mitochondria. If mitochondrial proteins decrease under respiro-fermentative metabolism this would cause lipid metabolism proteins to decrease, and why would an increase of mitochondrial proteins under purely respiratory metabolism not cause lipid metabolism proteins to increase?

Again, we did not intend to claim that this is something novel, but we just describe what our results show for the lipid metabolism related proteins. There may be some relationship between lipid metabolism proteins and mitochondrial proteins, but the emphasis point here is not this.

10. Line 186: More than mRNA fractions in lipid, glycolysis etc. being reversely proportional to growth rates (data ambiguous, statistics poor), it seems that these are positively correlated with specific metabolic states (respirative, respiro-fermentative, and fermentative). Please analyze both ways.

Response: The mRNA fractions allocated in lipid, glycolysis and chaperone are reversely proportional to specific growth rate, but not for all conditions. It showed that glycolysis and chaperon mRNAs fractions are reversely proportional to specific growth rate only under respirative state like their corresponding proteins fractions. While the lipid metabolism related mRNAs showed reversely proportional to specific growth rate only under respiro-fermentative condition. We do not have fermentative condition tested in our experiment.

11. Related to figure 3: the theoretical metabolic flux was estimated using GECKO and the genome-scale metabolic model of yeast. They correlate the flux with mRNA and protein levels separately, and look for reactions which show positive or negative correlation for both mRNA and protein. Some of this is quite contrived:

- In Line 216, they mention that TCA cycle & nucleotide biosynthesis show strong positive correlation for both mRNA and proteins. I cannot find a proper functional enrichment analysis for these, with significantly affected metabolic modules. Were by just looking at the metabolic map obtained from iPath3? While this might be so, it needs to be tested properly.

Response: Actually we did a thorough correlation analysis between reaction flux vs protein and mRNA levels, respectively. The corresponding python scripts and the depended data file are uploaded to github

https://github.com/bioexplore/multiomcispaperdata/tree/main/ResponsesToReviewers/Reviewer1/Q11_CorrelationBetweenFluxAndProteinAndmRNA), which include:

1. doCorrelationAnalysisFlux_mRNA_Protein.py python script that dealing with correlation analysis
2. FluxTranscriptomeAndProteomeCorrelationData.xlsx Excel file containing Yeast 7.6 genome scale metabolic model, absolute proteome data and absolute Transcriptome data

In short, we first extracted all genes whose coding proteins have metabolic function based on the Yeast7.6 model, then absolute quantification data of the corresponding proteins and mRNAs were utilized in the correlation analysis.

Gene-Protein-Reaction rule stored in the Yeast 7.6 model was used to pair reaction flux and corresponding protein or mRNA for doing the correlation. Accordingly, there are three conditions being considered in the correlation analysis: 1) one reaction has only one enzyme (unique protein or mRNA); 2) one reaction has several enzymes that do have the same function (isozymes, so several proteins or mRNAs for the reaction); 3) one reaction has one corresponding enzyme complex composed of several subunits (enzyme complex, so more than one protein together to have the function). For each condition, the correlation analysis applied different strategies: for the first condition, only one correlation coefficient was obtained for one reaction so nothing special; for the second condition, correlation coefficient between flux and the maximum abundant protein (or mRNA) was used as the final correlation coefficient for this reaction; for the last condition, correlation coefficient between flux and the minimum detected complex component (or mRNA) was used as the final correlation coefficient for this condition. Details are provided in the supplied python script.

With the proceeding steps, we obtained for each reaction two correlation coefficients: coeff_protein (correlation coefficient between flux and protein level) and coeff_mRNA (correlation coefficient between flux and mRNA level), the corresponding pval_protein and pval_mRNA were also obtained. Using $pval < 0.05$ as the statistics significance standard, we plotted the significantly positive correlation coefficient for both coeff_protein and coeff_mRNA (which is shown in Fig 3A with red rectangle border), and the significantly negative correlation points were emphasized in Fig. 3A with blue rectangle border.

With these significant points we then map these reactions into the iPath3 metabolic reaction network (which is shown in Fig. 3B) for better visualization of which pathway poses these significant correlated reactions. We have clarified this in the revised version of the manuscript. We also carried out a functional enrichment analysis for the significantly positive and negative correlation using DAVID, the result is shown below, and it gives the consistent results as we shown in Fig. 3B

(A)

(B)

Fig. R4 Enrichment analysis of significant correlation coefficient. (A) Positive correlation coefficient, total number of gene is 74; (B) Negative correlation coefficient, total number of gene is 40. Above each bar the number represent the adjust p-values by using the linear step-up method of Benjamini and Hochberg (1995).

- For Figure 3C, the authors can show simple bar graphs and present the interconnection between the multi-omics data. Currently, it's not readable at all.

Response: We supplied individual bar plot of Figure 3C in supplementary figures for better showing the contents.

12. How strong is the correlation between phosphorylation status of the enzyme and the metabolic flux? This is the weakest section of this manuscript. There are some mechanistic studies that look at phosphorylation of multiple metabolic enzymes, but no literature is cited that shows that increased phosphorylation leads to enzyme inhibition. Is there sufficient/any experimental evidence on this? If not, it is extremely premature for the authors to conclude this as a mechanism of how high flux is maintained despite enzyme concentrations.

Response: Many studies indeed showed that phosphorylation inhibits enzyme activity, we previously published a paper on inferring the functional phosphorylation event in *S. cerevisiae*, and it found that phosphorylation at T334, S336 or S338 of Elo2, S24 or S25 of Gpd1, Y309, S315 of Pda1, T117 or S119 of Pgm2, S240 of Sec53, S191, S192, S193 or S195 of Tps3 and T75, S77, S147, S155, S157 or S161 of Tsl1 inhibit activity (Chen et al, 2017). And finding that phosphorylation at T334, S336 or S338 of Elo2, S24, S25 or S27 of Gpd1, S72 of Gpd2 inhibit activity, have been experimentally validated previously (Lee et al., 2012; Oliveira et al., 2012; Olson et al., 2015). A rich source of the inhibition effect of phosphorylation on *S. cerevisiae* can be found in Table 1 of our previous review paper (Chen and Nielsen, 2016). We have made this clearer in the revised version of the manuscript.

References:

Chen Y., Wang Y.H., Nielsen, J. 2017, Systematic inference of functional phosphorylation events in yeast metabolism, *Bioinformatics*, 33(13):1995-2001.

Lee Y.J., Jeschke G.R., and et al., 2012, Reciprocal phosphorylation of yeast glycerol-3-phosphate dehydrogenases in adaptation to distinct types of stress, *Molecular and Cellular Biology*, 32(22):4705-4717.

Oliveira A.P., Ludwig C. and et al., 2012, Regulation of yeast central metabolism by enzyme phosphorylation, *Molecular Systems Biology*, 8: 623.

Olson D.K., Fröhlich F., and et al., 2015, Rom2-dependent phosphorylation of Elo2 controls the abundance of very long-chain fatty acids, *Journal of Biological Chemistry*, 290(7):4238-4247.

Chen Y., and Nielsen J., 2016, Flux control through protein phosphorylation in yeast, *FEMS Yeast Research*, 16(8): fow096.

13. The authors do not sufficiently explore the more obvious: the effect of metabolite concentrations in flux even at decreased enzyme concentrations, and how that responds to increasing growth rates. Further related to Fig 3C etc, on the specific activity of glycolytic enzymes or their usage, the authors systematically clarify then on how much of a 'buffer' of these enzymes are present? How much then is determined by mass action/substrate availability itself? Multiple studies (notably in yeast Hackett et al PMID: 27789812) quite clearly state that substrate concentrations are the primary drivers of metabolic reactions). So, in many ways the data in this paper primarily substantiates these findings systematically. There is no need to present this as novel, earth-shattering information.

Response: We do have done an exploration of the metabolite concentration effects on the reaction fluxes. Like what Hackett et al PMID: 27789812 did, we carried out a Bayes inference for the allosteric effects of metabolites on the central carbon metabolism enzymes. A detailed report interpreting what we did has been uploaded to github as supplementary file

https://github.com/bioexplore/multiomcispaperdata/tree/main/ResponsesToReviewers/Reviewer1/Q13_metabolite_effects):

1. Summary of Bayes inference results.docx

In short, we first give a detailed theoretical analysis of how we derive loglinear kinetic expressions. Then the derived kinetic models were used to form a Bayes inference objective function, and intrinsic turnover number of each metabolite to the reaction enzyme was proposed in this model. The proposed intrinsic turnover number tells how the corresponding metabolite affects the enzyme activity, i.e. a positive value means activation effect while a negative value means inhibition effect. Posterior distribution of the intrinsic turnover numbers for all tested metabolites was obtained by using Markov Chain Monte-Carlo based Bayes Inference.

However, only positive effect of ATP on phosphoglycerate kinase, positive effect of NADH on glyceraldehyde-3-phosphate dehydrogenase and positive effect of citrate on citrate synthase were identified by this analysis. Compared to the clear linear decrease of the glycolytic pathway enzyme fractions along with the specific growth rate, we do not think that the allosteric regulation effect play a key effect on the regulation of the fluxes, at least not for the EMP pathway.

14. In Fig 4D again the authors have presented R2. Not clear if the coefficient of correlation or the coefficient of determination for linear regression between amino acid biosynthesis intermediates and the specific growth rate is shown.

Response: Here in Fig. 4D, the Pearson correlation coefficient is used but not the coefficient of determination for the linear regression.

15. Also related to fig 4B and D: related lysine and arginine biosynthetic pathway genes with a positive correlation with growth rate. Can the authors see what the correlation of these pathway enzymes and transcripts, with ribosome biogenesis, across these broad growth rates/carbon availability? Recent experimental studies show that the synthesis and availability of lysine and arginine are the primary determinants of ribosome biogenesis (PMID: 33378328), apart from more extensive literature showing that the ribosomal proteins are extremely arginine and lysine rich. Of course, there is the very well established correlation of ribosome biogenesis with growth itself (from the Warner et al papers to more recent studies). So this seems like a much more plausible, biologically relevant correlation.

Response: To make clear whether lysine and arginine biosynthetic pathway proteins correlated well with that of the ribosomal proteins, we carried out a Spearman rank correlation analysis. The corresponding python script and dependent files are uploaded to github (https://github.com/bioexplore/multiomcispaperdata/tree/main/ResponsesToReviewers/Reviewer1/Q15_Lysine_and_Arginine_pathway_enzymes_correlation_with_ribosome_proteins), which include:

1. Lysine_Arginine_pathway_analysis.py Python script for doing the correlation analysis
2. Pvsm_new.xlsx Dependent file that store proteome data
3. eLIFE-ProteinCategoriesModify20200527.xlsx Proteome category file with the ribosome
4. Lysine_Arginine_biosynthesis_genes.xlsx Protein list that contain all get involved in lysine biosynthesis or arginine biosynthesis

5. `corrected_arginine_vs_ribosome_spearman_correlation.csv` Output correlation analysis results, in which FDR correction of the p-values were obtained
6. `Spearman_correlation_dist_lysinе_vs_ribosome.png` Histogram plot of the correlation coefficient distribution for lysine biosynthesis proteins
7. `Spearman_correlation_dist_arginine_vs_ribosome.png` Histogram plot of the correlation coefficient distribution for arginine biosynthesis proteins

The following two plots show the distribution of the Spearman correlation coefficient for lysine and arginine, respectively.

Fig. R5 Distribution of Spearman correlation coefficient for lysine biosynthetic proteins vs ribosomal proteins (a); and for arginine biosynthetic proteins vs ribosomal proteins (b)

The above plots show the distribution of the Spearman correlation coefficient carried out on biosynthetic proteins with ribosomal proteins for lysine and arginine, respectively. It can be seen that lysine biosynthetic proteins show much higher significant positive correlation with ribosomal proteins (over 40%), while arginine biosynthetic proteins show almost no significant correlation with ribosomal proteins. This cannot conclude that the proteins on the lysine and arginine biosynthetic pathways correlate with ribosome biogenesis. As respect to the reviewer recommended paper (PMID: 33378328), their study focused on the role of the transcriptional factor Gcn4, which is activated by methionine addition. They observed activation of 10/13 genes of arginine biosynthesis and 7/13 genes of lysine biosynthesis by Gcn4. However, their experimental set-up is quite different from our study: 1) they added methionine in the medium to activate Gcn4; 2) they only compared wild type with $\Delta gcn4$ with only glucose or glucose + methionine in the medium; 3) no Gcn4 protein was detected in our proteome data in all the 9 chemostat experiments. Together with the correlation analysis between biosynthetic pathway proteins and ribosomal proteins for lysine and arginine respectively, it can be seen that the biosynthesis of lysine seemed to be correlated to ribosomal proteins, and detailed check found that all significant correlations for lysine are mainly focused on Lys12, Lys20, and Lys4. Among these proteins, Lys20 catalyzes the first step of the *de novo* lysine biosynthesis.

As regard to the concern that lysine and arginine are the two most abundant amino acids in ribosomal proteins, we checked the amino acid composition and the following plot was obtained. In total, 134 ribosomal proteins were checked for their amino acid composition, and we indeed

found that the average two most abundant amino acid in the ribosome proteins were lysine (12%) and arginine (9%), but alanine also possessed 9% in average. Looking at the amino acid composition plot, it can be found that there is a large standard deviation for each amino acid. Which means that the composition of the ribosomal proteins varied a lot among the individual proteins. Corresponding python script, data and results are uploaded to github (https://github.com/bioexplore/multiomcispaperdata/tree/main/ResponsesToReviewers/Reviewer1/Q15b_Ribosome_amino_acid_statistics).

Fig. R6 Amino acid compositions of ribosomal proteins. Here the 134 ribosomal proteins detected in our proteome data were considered. The average percentage of each amino was calculated based on all 134 ribosome proteins sequences, standard errors were calculated based on individual protein.

Our analysis found that the amino acid biosynthesis pathway is not controlled by the protein level, but the enzyme usage correlated with the specific grow rate well.

Also related to Fig 4 (lines 357-62 etc.): “Increased enzyme concentration and enzyme saturation in amino acid metabolism further supports this flux increase for increasing growth rates.” This is fully consistent with Hackett et al 2004 and other studies. Also see your lines 395-398, which states the same point (without including these references).

Response: Yes, they are consistent, and corresponding reference has been added in the revision.

More minor points:

1. Line 351, in order to explain the cause behind low amino acid concentration despite high tRNA synthetase abundance, one needs to calculate the flux through the tRNA synthetase (which can be easily done using the metabolic model) and check if that correlates with the tRNA synthetase abundance.

Response: We picked up all tRNA synthetase fluxes simulated by the genome scale metabolic model, and extracted the corresponding abundances of the enzymes for these reactions from the

proteome data. We performed correlation analysis between the fluxes and enzyme abundances, and found that almost all tRNA synthetase abundances are correlated with the corresponding fluxes, except Ded81 and Dia4. We found that Dia4 has an isozyme Ses1, which means that it is Ses1 rather than Dia4 catalyzes the seryl-tRNA synthesis reaction. As regard to Ded81, we think that it may be also true that this enzyme correlated to the flux as the corrected p value is 0.052. Based on these checks, we conclude that Aminoacyl-tRNA biosynthesis could be mainly controlled by the enzyme protein levels.

Corresponding script and results are uploaded to github (https://github.com/bioexplore/multiomcispaperdata/tree/main/ResponsesToReviewers/Reviewer1/QS1_Correlation_between_protein_and_aminotacyl-tRNA_biosynthesis), which include:

1. Correlation_between_protein_and_tRNAsynthesis.py Script for doing the correlation analysis.
2. tRNA_Fluxes_proteome.xlsx Input data for the correlation analysis.
3. Corrected_pvsflux_correlation.datemark.csv Correlation analysis results and corrected p values.

2. There is some slightly careless referencing. For example, refs 6 and 7 are hardly illustrative of the use of yeast as “a model organism for deciphering molecular mechanisms in cellular and molecular biology”, but this made me smile. Please go through the references carefully, and appropriately cite past literature.

Response: Thanks, we have checked the references carefully and corrected the issue. We also added new reference in the introduction part to make the description clearer and more relevant.

3. Line 139: There is no strong evidence (or backing references, in lieu of evidence) with which to state that decrease in glycolytic and chaperon fraction is strategically important for making space for other proteins.

Response: We rephrased to “the decrease in glycolytic and chaperon fraction accompanied by increase of other groups of proteins, like mitochondrial proteins, translational proteins, etc.”

3. Figure 4A and 4C are noisy and very difficult to read. Can the authors present only the enzymes which show significant changes with respect to the independent variable? The rest can be a supplemental

Response: Just did as what the reviewer recommended.

4. Several typos in the manuscript. Please proof carefully. For example, correct spelling of slope (slop in places), line 227 “...important role to fulfill...” and many, many more

Response: We checked the manuscript carefully, and corrected all typos.

Reviewer #2 (Remarks to the Author):

Xia et al. describe a study in which they cultured yeast in chemostats with glucose as the sole carbon source. They used 9 different dilution rates that equal the specific growth rate (ranging

from 0.025 per h to 0.4 per h) and measured the transcriptome, proteome, phosphoproteome and metabolome. Additionally, the authors inferred fluxes from measured exchange fluxes of glucose, ethanol, O₂ and CO₂. Main findings are: i) the concentration of many proteins (or groups of proteins like ribosomal proteins) is growth rate dependent, ii) phosphorylation of glycolytic enzymes and chaperons is growth rate dependent and iii) proteome allocation is determined at the transcriptome level. The strength of the work is the well-designed experiment that employs chemostats to enforce different growth rates and different metabolic states (respiratory vs respiro-fermentative metabolism) in yeast. The data set is impressive, of high quality and probably one of the most comprehensive multi-omics data sets. My main concern is that the analysis of the data is very descriptive and focuses mostly on correlations between growth and the different “omics” levels. Specific points are:

1) A large number of studies examined growth-rate dependencies of transcription, translation and metabolism and the authors do a great job explaining them in their introduction. However, the manuscript should better describe which growth-rate dependencies have been observed by others (and in which organism) and which growth rate effects are newly revealed by their data. For example, Metzler-Raz et al (Ref [8] in the text) observed similar relationships in yeast, but it is unclear if there are differences to the data shown here and if they are due to the different conditions.

Response: Thanks for the kind suggestion. We have done a thorough reading on this issue and updated our introduction accordingly. To clarify the differences to the data shown in this paper and that in Metzler-Raz et al and other publications, text was added in the Discussion part of the revised manuscript.

2) In Figure 1 C and D the authors fit two linear functions to one proteome category. What is the basis to assign data points to the first or the second function (e.g. in Fig 1C there are 3 data points in the second fit and in Figure 1D there are 4 data points in the second fit)? From Figure 1 B it seems that only the two highest growth rates are in the respiro-fermentative regime.

Response: We also noticed the difference when we did the analysis but we did not mention this in the previous version of the manuscript. If we look at Fig. 1B, which showed specific rates of several extracellular metabolites (glucose, ethanol, oxygen and carbon dioxide), it showed that the turning point for glucose and ethanol specific rates was at the specific growth rate of 0.28 h⁻¹. This is consistent with reports of van Hoek et al, Appl. Environ. Microbiol., 1998 (PMID: 9797269) that studied on *S. cerevisiae* DS 28911 in glucose-limited chemostat. However, if we looked at the respiration (the specific oxygen uptake rate and specific carbon dioxide production rate) profiles, it seemed to be at $\mu = 0.254\text{h}^{-1}$ (the last but three point) where the two respiratory specific rates (q_{o_2} and q_{co_2}) decoupled. An early Crabtree effect chemostat study on *S. cerevisiae* CBS 8066 (Fig. 2 in PMID: 2566299) also showed the same early change on q_{o_2}/q_{co_2} (when $\mu < 0.25\text{h}^{-1}$) before ethanol was observed to accumulate. In this study there was also observed a decoupling of q_{o_2} and q_{co_2} , before the specific growth rate reached the critical value where ethanol started to accumulate.

Here, the critical specific growth rate was defined as when ethanol begins to accumulate, i.e. at 0.28 h⁻¹, however, as mentioned q_{o_2} and q_{co_2} decouple before the specific growth rate reaches 0.28 h⁻¹. This may be the reason for the best fit curves difference for Fig. 1C and Fig. 1D. But why q_{o_2}

and q_{co2} decouple below the critical specific growth rate is unclear. We added a discussion of this in the revised manuscript.

3) The data should contain information about the regulatory mechanisms that lead to the changes in the proteome fraction. Since the authors nicely show that the changes occur at the transcriptional level, they could go one step further and uncover the transcriptional regulators that are involved. For example, which regulator could be responsible for downregulating glycolysis enzymes? At least the authors can discuss some hypothesis about this.

Response: Transcriptional regulation in yeast has been extensively studied earlier and several different transcription factors have been identified, e.g. Gcr1 and Gcr2, but also the stress response transcription factors Msn2 and Msn4 regulate glycolytic genes. Functions of these TFs have been discussed by Holland et al. 2019, *Nucleic Acids Research*, (PMID: 30976803).

Additionally, we searched the literature and found a paper by Sierkstra et al. in 1992 (PMID: 1487726) describing the transcription level of glycolytic genes in glucose-limited chemostats with dilution rate ranging from 0.05 to 0.315 h⁻¹. But they found that most of the mRNA levels of glycolytic enzymes remained constant, while they observed that enzyme activity decrease, e.g. of phosphoglucomutase, so they concluded that there is no transcriptional or translational regulation of glycolytic flux.

However, we still want to see whether some transcription factors take the responsibility to regulate the expression level of glycolytic enzymes. Based on the reviewer's comment, we checked the reported transcription factors by Lee et al., 2002, *Science* (PMID: 12399584) in *S. cerevisiae*, and we selected 28 TF regulators and their target-genes confirmed by gene specific PCR (Table S1 in PMID: 12399584). However, we did not find any transcription factor having their target gene to be among the glycolytic genes.

Another paper by Ferreira et al., 2007 (PMID: 17035097) predicted hypoxia response elements (HRE) carrying genes, which are target of hypoxia-inducible factor-1 (HIF-1, a human transcription factor), that can cause promoted glycolytic flux in tumor cells. The authors hypothesized that *S. cerevisiae* possess an analog of HIF-1. In their predicted HRE carrying gene list, we found TDH2 (coding glyceraldehyde 3-phosphate dehydrogenase), which is a glycolytic gene. Even though the HIF-1 analog TF may be really true in *S. cerevisiae*, only one target gene in the glycolytic pathway cannot explain our observations.

Szatkowska et al. (doi: 10.1042/BCJ20180701) showed that RNA polymerase III and its negative regulator Maf1 can regulate the glycolytic flux. And they found that glycolytic enzymes abundance will decrease (2- and 2.6-fold) when RNAP III is compromised, especially Tdh1, Tdh2, Eno1, Pfk1 and Glk1. We checked our proteome data for the proteins of RNAP III and Maf1, however, none of these proteins were included in the proteome data. Thus, we checked the mRNA levels of these genes instead. The following two plots show how these mRNA level change along with specific growth rate.

Fig. R7 mRNA expression level for RNA polymerase III (a) and its negative regulator Maf1 (b).

Based on these data, we observed downregulation of Maf1 on the mRNA level along with specific growth rate, while its negatively regulated target the RNA polymerase III, which keeps its components constant on the mRNA level except Tfc6 (two-fold decrease) along with the specific growth rate. We hypothesize that downregulated Maf1 level release its negative regulation of RNA polymerase III activity, however, sharp decrease of Tfc6, component of RNAP III, causes the decrease of most glycolysis proteins, like observed in Szatkowska et al. (doi: 10.1042/BCJ20180701).

We have added a discussion comment about this in the manuscript and detail discussion is given in supplementary text.

References:

- Sierkstra, L.N., J.M.A. Verbakel, and C.T. Verrips, Analysis of transcription and translation of glycolysis enzymes in glucose-limited continuous cultures of *Saccharomyces cerevisiae*. *Journal of General Microbiology*. 138(1992) 2559-2566.
- Lee, T.I., N.J. Rinaldi, F. Robert, D.T. Odom, Z. Bar-Joseph, G.K. Gerber, N.M. Hannett, C.T. Harbison, C.M. Thompson, I. Simon, J. Zeitlinger, E.G. Jennings, H.L. Murray, D.B. Gordon, B. Ren, J.J. Wyrick, J.-B. Tagne, T.L. Volkert, E. Fraenkel, D.K. Gifford, and R.A. Young, Transcriptional regulatory networks in *Saccharomyces cerevisiae*. *Science*. 298 (2002) 799-804.
- Ferreira, T.C., L. Hertzberg, M. Gassmann, and É.G. Campos, The yeast genome may harbor hypoxia response elements (HRE). *Comparative Biochemistry and Physiology Part C: Toxicology & Pharmacology*. 146(1) (2007) 255-263.
- Szatkowska, R., M. Garcia-Albornoz, K. Roszkowska, S.W. Holman, E. Furmanek, S.J. Hubbard, R.J. Beynon, and M. Adamczyk, Glycolytic flux in *Saccharomyces cerevisiae* is dependent on RNA polymerase III and its negative regulator Maf1 *Biochemical Journal*. 476(6) (2019) 1053-1082.

4) It is unclear if the authors considered growth-rate dependency of cell volume. For many organisms the cell volume changes with the growth rate. If such cell volume changes also occur in yeast they can substantially impact the results.

Response: Cell volumes at different growth rates may be different under some conditions, and it will affect the absolute concentration of individual transcript or protein. However, the main conclusion of our study is based on relative abundance, i.e. mass fractions, for example, the linear

relationships between fraction of each protein group and specific growth rate (Fig. 1C and 1D). So that the cell volume will not have any effect on our conclusion. Furthermore, according to the study of Boer et al., 2010, Mol. Biol. Cell, 21(1): 198-211 (PMID: 19889834), in which five nutrient limitation chemostats including glucose limitation chemostat, cell volume does not change a lot across a wide range of specific growth rate (from 0.05 h⁻¹ to 0.3 h⁻¹, covering both respiratory and respiro-fermentative conditions).

5) The authors could use a more systematic approach to integrate the different omics levels in order to extract the full information from their data set. The insights from the current analysis reflect to a large part the findings in *E. coli* (Scott et al., Reference [11] in the text) and yeast (Hackett et al., [13]). For example, in line 280 hierarchical regulation analysis is mentioned, which is indeed a great tool to integrate all data types and test how the different levels contribute to flux regulation. But the analysis was restricted to i) only the proteome and ii) only the central metabolism and amino acid synthesis. Did the authors attempt to estimate the coefficients of all enzymes, metabolites and PTMs (as they mention themselves in line 281)?

Response: Scott et al. reported a growth law empirical model to interpret the linear relationship between cell growth rates with translation capacity of the *E. coli* cell. Their findings of the growth law do make sense. However, they did not report how the other functional category of the proteome will cooperate with the translation proteome demand when cells grow faster, as they simply combined all other groups into a non-translation proteome. Instead, we extended the relationship between proteome and cell growth rate to more detailed functional groups besides the translation category.

Hackett et al. did a great job on the systematics level to decipher what controlled the reaction fluxes for *S. cerevisiae* under a wide range of different steady state conditions. Their work focused mainly on metabolic reactions, and they concluded that intracellular metabolite concentrations are the strongest driver for determining reaction rates. However, many other functional parts of the cell like translation, chaperone functions, and transcription factors were omitted in their study, as they are not a specific biochemical reaction. Instead, we aimed to obtain a broader view by investigating how yeast cells allocate its resources to different functional groups on both the transcriptome and the proteome level. We provide new findings on yeast physiology under a wide range of specific growth rates. We actually also carried out a Bayes inference for studying the allosteric effects of metabolites like Hackett et al. did but with a loglinear kinetics equation rather than the Michaelis like kinetics. Details of this analysis is attached as a single document and uploaded to github (<https://github.com/bioexplore/multiomcispaperdata/tree/main/ResponsesToReviewers/Reviewer2>):

1. Summary of Bayes inference results.docx

6) In the last part of the model it is not clear how and why enzyme usage was calculated with the GECKO method. Enzyme usage can be directly estimated for each enzyme individually using fluxes, kcat and the proteome data. The authors should discuss the advantage/difference of the GECKO method.

Response: GECKO is based on the FBAwMC approach (Beg et al, 2007) but extended to limit each individual enzyme, thereby giving a physiologically constrained and thus more feasible solution. By comparing enzyme measurements to their usage in the model, we could compute enzyme usage percentages, which can be interpreted as a new layer of information connecting proteomics and fluxomics, and can be studied to find usage trends among different experimental conditions. Moreover, by having several experimental conditions, we could find enzymes that are highly used among all conditions, which could be a sign of transcriptional regulation. Quote from our previous published paper by Sanchez et al., 2017. As advantage/difference of the GECKO method have been discussed in the original paper, we will not repeat this here in this work.

References

Beg QK, et al., Intracellular crowding defines the mode and sequence of substrate uptake by *Escherichia coli* and constrains its metabolic activity. *Proc Natl Acad Sci USA* 104: 12663 – 12668, (2007).

Sanchez BJ, Zhang C, Nilsson A, Lahtvee PJ, Kerkhoven EJ, Nielsen J. Improving the phenotype predictions of a yeast genome-scale metabolic model by incorporating enzymatic constraints. *Mol Syst Biol* 13, 935 (2017).

7) In the discussion the authors claim they “mapped the multilayer regulation structure in protein allocation”. I don’t agree with this statement because the authors only show correlations between the different layers but don’t infer regulatory interactions or show which mechanisms are active. For example, the decreasing concentrations of amino acids is an interesting finding (line 359), but the authors cannot show if they are affecting the respective feedback mechanism and at which layer (enzyme activity or enzyme concentration).

Response: We have revised the corresponding discussion part to make it clear that the conclusion drawn is supported by our data and analysis.

Reviewer #3 (Remarks to the Author):

Overview

Xia and coauthors provide a detailed multiomic characterization of *S. cerevisiae* growth across a range of specific growth rates in carbon-limited chemostats. Faster growth requires cells to make a similar amount of biomass in less time, and also encompasses the well known Crabtree effect. As a result, understanding the regulatory processes operating across this regime is of considerable interest. The experimental methodology and informatics used by the authors appears to be sound, and they report three major findings. First, separating proteins and transcripts into major cellular processes suggests that cells’ investment in these different categories is similar at the transcriptional and protein level. Thus, it is arguable that gross regulation of protein abundance is primarily at the transcriptional level. Second, increased activity of chaperones and glycolytic enzymes is consistent with a decrease in phosphorylation with specific growth rate. Third, increased amino acid biosynthesis can be partially explained by increased enzyme saturation. Of these findings, the regulation of glycolytic flux by dephosphorylation is arguably the most

important result but I have my doubts about the validity of this finding since metabolic regulation was not considered as a competing hypothesis. Interpretation of the results is also made more difficult by relatively poor writing and unpolished figures.

Major

1. The author's assertion that glycolytic flux control occurs primarily through phosphorylation ignores the metabolomics data that they have collected. Boer et al. 2010 "Growth-Limiting Intracellular Metabolites in Yeast Growing under Diverse Nutrient Limitations." Molecular Biology of the Cell indicates that upper glycolytic intermediates increase with specific growth rate in carbon limited yeast chemostats. To wit, increases in substrate saturation likely contribute to growth-rate dependent elevation of glycolytic flux. The authors should examine the influence of hierarchical control versus metabolic control in glycolysis simultaneously as they did for amino acid synthesis.

Response: The metabolic control cannot be directly estimated by hierarchical regulation analysis as we do not know the function of the metabolic part in the equation used for this analysis. In Boer et al. 2010, they showed that pyruvate can be a representative metabolite to stand for glucose limitation, as it increases with the specific growth rate. However, in our metabolite data for glycolysis intermediates this is not the case (shown below in Fig. R8).

Fig. R8 Relative abundance of glycolysis pathway intermediates for the nine dilution rates. In the plot D1 to D9 are in the ascending order, with D1 the minimal at $\sim 0.025 \text{ h}^{-1}$, and D9 the maximal at $\sim 0.4 \text{ h}^{-1}$.

Instead of doing hierarchical regulation analysis with the metabolite data, we did a kinetics study using Bayes inference to see which metabolite show noticeable allosteric effects on glycolytic

enzymes, like what Hackett et al. (PMID: 27789812) did. And the Bayes inference results showed only positive effect of ATP on PGK, positive effect of NADH on GAPD and positive effect of citrate on CITS. Details of the report are attached as a single document:

1. Summary of Bayes inference results.docx

2. Comparing “fraction of the proteome” to “fraction of the transcriptome” to make the argument that “proteome allocation pattern is mainly determined at the mRNA level” is difficult. To make this point the authors should directly compare these quantities rather than relying on readers comparing Figures 1 and 2. For example, if their assertion is true, the ratio of proteome fraction to transcript fraction should be independent of specific growth rate. Alternatively, a bivariate scatter plot could demonstrate proportionality.

Response: Thanks for this very good comment. We did a Pearson correlation analysis between proteome group fraction and transcriptome group fraction for individual functional categories. A python script and related dataset and results are uploaded to github (https://github.com/bioexplore/multiomcispaperdata/tree/main/ResponsesToReviewers/Reviewer3/Q2_Pearson%20correlation%20analysis%20functional%20groups%20level), which include:

1. Correlation_analysis_between_proteome_and_transcriptome_group_fractions.py
2. Group_comparison_between_proeome_and_transcriptome.xlsx
3. Corrected_pvsm_group_fraction_correlation_analysis_datemark.csv

The correlation results were shown in the following table, and correlation between proteins and mRNAs fraction in individual groups was plotted in Fig. R9

Table R1. Pearson correlation coefficient and corrected pvalue with FDR < 0.05

Category	Coef	pvalue	pcorr	FDR<0.05
Translation	0.967	2.14E-05	4.49E-05	TRUE
Glycolysis	0.953	7.00E-05	0.00011	TRUE
Mitochondria	0.870	0.002291	0.002406	TRUE
AminoAcids	0.968	1.80E-05	4.49E-05	TRUE
Lipid	0.936	0.000206	0.00026	TRUE
Chaperone	0.975	7.63E-06	4.49E-05	TRUE

Fig. R9 Linear relationship between proteome fraction and transcriptome fraction in individual functional group.

The results of Table R1 and Fig. R9 show that at the functional group level the transcriptome fractions are consistent with that of the proteome fractions. Which means that *S. cerevisiae* allocate its resources based on functional groups for organizing its transcriptome and proteome. However, if we look Fig. R9 carefully we can see that the fraction assigned at mRNA level for each functional group are lower than those assigned at the protein level except for the mitochondria group.

3. The paper requires thorough proofreading. A subset of the word usage, formatting and major English errors can be found in minor.

Response: Thanks, we edited the manuscript thoroughly.

Minor

1. In several instances the authors report counts of genome-scale measurement p-values less than a cutoff without applying false-discovery rate control. For example, “We filtered out significant (p value < 0.05)”.

Response: We did multiple testing FDR correction with $FDR < 0.05$, and corrected p values were obtained and used as the filtering rule (corrected p value < 0.05). The corresponding script for dealing with the issue is attached, codes in the script were used also for generating the updated version of Fig. 3A. The script is uploaded to github (https://github.com/bioexplore/multiomcispaperdata/tree/main/ResponsesToReviewers/Reviewer3/QS1_Redo_correlation_analysis_for_Fig3A):

1. doCorrelationAnalysisFlux_mRNA_Protein.py Python script for doing correlation analysis between reaction flux and corresponding mRNA abundance, and that between reaction flux

and corresponding protein abundance.

2. To make the case that the levels of unphosphorylated glycolytic enzyme increases with specific growth rate more compelling, the authors should add this information to Figure 3C. The levels of the total enzyme and phospho enzyme look really similar, which would mean that the phosphorylation events could be totally inert.

Response: The unphosphorylated proteins cannot be estimated as the phosphoproteomics data are not absolute. To make it clear all individual plots including both proteins and phosphoproteins plots were added in the supplementary text as Fig. S8B and S8C. These figures clearly show the difference between protein level and phosphorylation level for individual proteins.

3. The author's overstate "Both hierarchical regulation analysis and FPE inferring confirmed function inactivation due to phosphorylation of seven of the glycolysis enzymes" as well as "We therefore conclude that phosphorylation of both glycolytic enzymes and chaperones seems likely to reduce their activities". Based on my reading of FPE, it is primarily picking up on the anticorrelation of flux and phosphorylation/phospho-protein levels. Unless competing hypotheses can be ruled out (see major 2), it's more appropriate to indicate that increased flux could be caused by [or, is consistent with] a decrease in inhibitory phosphorylation events.

Response: For these glycolytic enzymes, we can rule out the control by protein levels and metabolite levels based on our analysis, and we indeed observed the anticorrelation of flux and phosphorylation levels. However, we are not able to measure the unphosphorylated parts. Therefore, as suggested, we have changed our statement in the manuscript as "the increased glycolytic flux could be caused by the decrease in inhibitory phosphorylation events"

4. The authors should justify why their analysis of protein phosphorylation occurs at the protein-level rather than site-level. Some modifications may affect catalysis while others would not. I can imagine the real reason here is that we don't know the role of most phosphosites, and like in conventional proteomics, the phosphopeptides we obtain are just a sample of the total set. This is reasonable, but caveats should be spelled out.

Response: We do have site-level phosphorylation event analysis, but the results are not shown in the manuscript. We now added one paragraph that provide details of the results of the site-level phosphorylation events.

5. Please provide additional details on Figure S8. It is unclear why the role of hierarchical control seems to fluctuate so much as a function of specific growth rate when most measures vary smoothly.

Response: Sorry for the unclarity in Figure S8. Actually, the legend (located in the right middle of the plot) in Fig. S8A provides some of the information. We carried out nine chemostats with the dilution rate ranging from 0.025 to 0.4 h⁻¹, corresponding to d1 (0.027 h⁻¹), d2 (0.044 h⁻¹), ..., d9 (0.379 h⁻¹). For doing the hierarchical regulation analysis we needed two steady states data, so we choose two consecutive chemostats for doing the hierarchical regulation analysis, e.g. d2 vs d1, d3 vs d2, ..., d9 vs d8. This is why we have 8 regulation coefficients (8 blocks for each enzyme) in the figure. As we choose two consecutive chemostats but not one specific chemostat as the unique

reference for all the regulation coefficient calculation, this may be the reason why we got fluctuating regulation coefficients.

6. Describing whether the previously reported “phosphorylation events on Pfk2, Fba1 and Gpm1” were also found to be inhibitory by Oliveira et al. would more strongly support the findings of this study.

Response: Oliveira et al. reported inhibition by phosphorylation of Pfk2 at S163, and we also found phosphorylation events of Pfk2 inhibit enzyme activity at S163 (however statistically non-significant) but we also found another inhibition site (T152) and one activate site (S160) (Shown in supplementary figure Fig. S7). Oliveira et al. also reported the S313 site of Fba1 with an activation effect but we inferred no significant effects at the same site of Fba1 (Fig. S7). Inhibition function of Y90 for Gpm1 was observed in this work but not reported in Oliveira et al. In addition, we also observed inhibition function of phosphorylation events for Pfk1 T178, S179, S185, S188, S189, and S192, for Tdh1 S149, Tdh2 S149, Tdh3 S149, Pgk1 Y75, Eno1 Y259, and Eno2 Y259. All these results and the functional phosphorylation events analysis results were added in the revised supplementary files.

7. Consider specifying that the slope from the RNAseq, absolute mRNA regression was used to estimate the absolute mRNA for the remaining genes. Mentioning the R2 statistic then that “The same correlation was applied” may be confusing.

Response: The standard curve between mRNA absolute concentration and FPKM value were obtained for 18 mRNAs (whose FPKM values cover over 80 % of the dynamic range of mRNA expression FPKM values at reference conditions) with their FPKM values and absolute concentrations measured at the same condition. The corresponding data are listed in Table S2. The standard line obtained is plotted as follows,

Fig. R10. Regression line between log10 mRNA absolute concentration and its log10 FPKM value of the 18 selected mRNAs.

Then with all other mRNAs' FPKM value obtained using Cufflinks 2, their absolute concentrations at the reference condition were calculated based on the regressed standard curve. The absolute concentrations under other conditions were calculated using fold-change values of individual genes as respect to the reference condition combined with their absolute concentration values under reference conditions.

Figure 1C doesn't look like an R2 of 1, how many significant digits are used?

Response: In Fig. 1C, for translation proteins the regressed R2 is 0.99, mitochondrial proteins R2=0.98, and amino acid biosynthesis proteins R2=0.88.

8. In 1C/D, "proportionality" is inappropriate; in these instances correlation would be better.

Response: "proportional" was replaced with "correlated with".

9. To make Figure 3 cleaner the authors should consider increasing the resolution of the legend text in 3A/B and removing unnecessary annotations on reaction map (3B) and axis text (3C).

Response: We have done as the reviewer suggested. In addition, we supplied a detailed individual plots shown in Fig. 3C in supplementary figures.

10. "majority of the dry cell weight" implies > 50% of the total, while protein ranges from 30-50%.

Response: "Majority" was replaced with "large part".

11. English, grammar: "P values were gotten from", "two groups increased indeed linearly", "these two groups had a profile that was metabolism depending", "abundance of proteins are 3 orders bigger", "the functional parts of the life process", "Enzyme saturation play an important role fulfil amino acid biosynthesis", "We are confident that these findings will besides providing abasic insight into regulation of metabolism in eukaryal model organism S. cerevisiae it will also assist..."

Response: All the above mentioned grammar issues were corrected in the revised version.

12. Formatting, missing characters: "-20 until"

Response: Missing character "°C" were added.

13. Formatting, typos: "fulfil", "slop", "underlyingprinciples", "2302fluxes", "Mitochodira", "Chaperons" (Figure S5)

Response: All these issues were fixed in the revised version.

14. Formatting, excess whitespace: "volume before"

Response: The excess whitespace was deleted.

REVIEWERS' COMMENTS

Reviewer #1 (Remarks to the Author):

This revised version of the manuscript is outstanding. The authors have done a fantastic job of taking into regard all the reviewer's comments, and addressed every one of them. The reanalysis of some of the data, the addition of key controls (especially in the statistical analysis and interpretations therein), and the added figures/supplemental have substantially improved the clarity of the manuscript. The changes in text now make the article more readable, and the needless over interpretations of data have been suitably toned down.

While I might quibble with some interpretations of the data, the data themselves are presented unambiguously. The findings are novel, and the data sets fantastic. I would strongly recommend that this outstanding article be published without any additional requirements.

Reviewer #3 (Remarks to the Author):

The authors present a substantially revised manuscript which addresses many of my past scientific concerns. However, the authors have scarcely addressed my major 3 comment that the manuscript needs through proofreading. Poor english and writing should not raise to the level of a major concern that reviewers must flag as the authors are fully able to address such shortcomings. This concern was scarcely addressed in the revision rather the specific typos and english errors that I called out were fixed but a slew of new and previously existing writing issues remain. I tried to re-read the manuscript to see how it holds together with the many additions that were made to satisfy my and the other reviewers concerns but the poor writing was too distracting to dig into this manuscript's complex science. In its current state this manuscript is not suitable for a general audience particularly not in a journal with the title "Communications".

REVIEWER COMMENTS and OUR RESPONSES

Reviewer #1 (Remarks to the Author):

This revised version of the manuscript is outstanding. The authors have done a fantastic job of taking into regard all the reviewer's comments, and addressed every one of them. The reanalysis of some of the data, the addition of key controls (especially in the statistical analysis and interpretations therein), and the added figures/supplemental have substantially improved the clarity of the manuscript. The changes in text now make the article more readable, and the needless over interpretations of data have been suitably toned down.

While I might quibble with some interpretations of the data, the data themselves are presented unambiguously. The findings are novel, and the data sets fantastic. I would strongly recommend that this outstanding article be published without any additional requirements.

Response: We thank the reviewer for all your wonderful comments, with which we improved the manuscript a lot and without your help we can not make it. Thanks a lot.

Reviewer #3 (Remarks to the Author):

The authors present a substantially revised manuscript which addresses many of my past scientific concerns. However, the authors have scarcely addressed my major 3 comment that the manuscript needs through proofreading. Poor english and writing should not raise to the level of a major concern that reviewers must flag as the authors are fully able to address such shortcomings. This concern was scarcely addressed in the revision rather the specific typos and english errors that I called out were fixed but a slew of new and previously existing writing issues remain. I tried to re-read the manuscript to see how it holds together with the many additions that were made to satisfy my and the other reviewers concerns but the poor writing was too distracting to dig into this manuscript's complex science. In its current state this manuscript is not suitable for a general audience particularly not in a journal with the title "Communications".

Response: Thanks for pointing out the language issue. To correct this issue, we turned

to Nature Research Editing Service for help. After their editing we think the manuscript is now much clear and should be suitable for consideration of publication. We still want to thank the reviewer for your help to improve the quality of the work.